



# Evaluation of PMIP2 and PMIP3 simulations of mid-Holocene climate in the Indo-Pacific, Australasian and Southern Ocean regions

Duncan  Ackerley[1], Jessica Reeves[2], Cameron Barr[3,4], Helen Bostock[5], Kathryn Fitzsimmons[6], Michael-Shawn Fletcher[7], Chris Gouramanis[8], Helen McGregor[9], Scott Mooney[10], Steven Phipps[11], John Tibby[3,4], and Jonathan Tyler[3,4]

[1]ARC Centre of Excellence for Climate System Science, School of Earth, Atmosphere and Environment, Monash University, Victoria 3800, Australia
[2]Federation University, Faculty of Science and Technology, Mt Helen, Ballarat, Victoria 3353, Australia
[3]Department of Geography, Environment and Population. University of Adelaide, North Terrace, Adelaide, 5005.
[4]Sprigg Geobiology Centre. University of Adelaide, North Terrace, Adelaide, 5005.
[5]National Institute of Water and Atmospheric Research, 301 Evans Bay Parade, Greta Point, Wellington, New Zealand.
[6]Max Planck Research Group for Terrestrial Palaeoclimates, Climate Geochemistry, Max Planck Institute for Chemistry, Hahn-Meitner-Weg 1, 55128 Mainz, Germany
[7]School of Geography, University of Melbourne, Parkville, Victoria, Australia, 3010.
[8]Department of Geography, National University Of Singapore, 10 Kent Ridge Crescent, Singapore 119260
[9]School of Earth and Environmental Sciences, University of Wollongong, Northfields Ave, Wollongong NSW 2522 Australia
[10]School of Biological, Earth and Environmental Science, UNSW, Sydney 2052.
[11]Institute for Marine and Antarctic Studies, University of Tasmania, Hobart, Tasmania, Australia

*Correspondence to:* Duncan Ackerley (duncan.ackerley@monash.edu)

**Abstract.** Paleoclimate proxy reconstruction initiatives, such as the Australian component of the international paleoclimate synthesis effort: INTegration of Ice core, MArine and Terrestrial records (OZ-INTIMATE), are important as they provide evidence of past climatic conditions that are necessary to evaluate global General Circulation Models (GCMs). One of the key outputs from the OZ-INTIMATE project was the production of spatially-coherent, climatic reconstructions over the southern

5    Maritime Continent, Australasia and the Southern Ocean. The OZ-INTIMATE results were presented as regional, "simplified patterns of temperature and effective precipitation" and those regions spanned a large enough area to contain several GCM grid boxes. Therefore, the "upscaling" of individual reconstructions (through OZ-INTIMATE) to a scale that was resolved by GCMs, presented an ideal opportunity for a direct comparison.

This study uses the same "simplified patterns of temperature and effective precipitation" approach from OZ-INTIMATE on

10   data from an ensemble of GCMs. The GCM data are taken from the Paleoclimate Modeling Intercomparison Project (PMIP) mid-Holocene (6000 years before present, 6 ka) and pre-industrial control (c1750 C.E., 0 ka) experiments. The synthesis presented here shows that, on the whole, the models and proxies agree on the differences in climate state for 6 ka relative to 0 ka, when they are insolation driven. The main disagreement between the models and proxies occurs over the Tropical West Pacific warm pool and arises from an intensification of an existing error (the "cold tongue bias"). This study also presents a mechanism

15   whereby the strength of the Southern Hemisphere, mid-latitude westerly wind strength reduces but rainfall increases over the





southern temperate zone of Australia. Such a mechanism may be useful for resolving disparities between different regional proxy records, and model simulations. Finally, after assessing the available datasets (model and proxy), opportunities for better model-proxy integrated research are presented.

## 1 Introduction

Paleoclimate proxy reconstructions provide evidence of past climate, which is vital for giving context to projections of future anthropogenic climate change. Moreover, proxies give an observationally based dataset spanning centuries to millennia against which simulations of future climate (performed by General Circulation Models, GCMs) can be compared. A direct comparison between model and proxy data is difficult, particularly when GCM grid spacing is ∼100 km (or more) and proxies represent climatic information at a specific place or region. A method of bridging this scale gap is to "upscale" various regionally-

coherent proxy reconstructions to a scale that is resolved by GCMs. Conversely, large-scale GCM data can be "downscaled" using known circulation characteristics to provide local-scale (≤100 km) estimates of climatic variables (e.g. precipitation and temperature). Such an upscaling approach was adopted in Lorrey et al. (2007, 2008) to use regionally coherent climate proxy data (temperature and precipitation) to infer circulation characteristics over New Zealand. Subsequently, Ackerley et al. (2011) applied the reverse method to downscale coarse-resolution GCM data to infer regional temperature and precipitation

characteristics over New Zealand, which provided a platform to evaluate the merits of both model and proxy datasets. Over Australasia, the Maritime Continent and Southern Ocean, such an upscaling/downscaling approach has not yet been attempted to integrate proxy and model data; however, the synopsis of the OZ-INTIMATE initiative Reeves et al. (2013a) presents an innovative way to do so, which is explored here.

The Southern Ocean, Australasian and, southern Maritime Continent region considered by OZ-INTIMATE spans from

10°N to 60°S and 115°E to 155°E (Fig. 1). The region incorporates the tropical (including the Sunda Shelf), arid, temperate and southern Indian Ocean/Southern Ocean climatic zones (see Fig. 1). While individual climate reconstructions for the last 35,000 years are discussed individually elsewhere (see Bostock et al., 2013; Fitzsimmons et al., 2013; Petherick et al., 2013; Reeves et al., 2013a, b, for more details), it is the schematic representation of "simplified patterns of temperature and effective precipitation between regions" (p24 of Reeves et al., 2013a, their Figure 4) that provides the opportunity for direct model-

proxy comparison. Hence we are applying a synthesis model and refer only to specific records for validation or discussion of discrepancy. We have also included reference to some relevant records that have been published since the OZ-INTIMATE compilation, which are included in the discussion below.

Reeves et al. (2013a) sub-divide the Australasian-southern Maritime Continent region into four main zones, which are indicated by the solid lines in Fig. 1 tropical, sub-tropical, temperate and Southern Ocean. Reeves et al. (2013a) also sub-divide

those four regions into the tropical north-west and north-east (perennially wet equatorial), tropical south-west and south-east (monsoon region), the arid sub-tropics (continental interior of Australia), the temperate east and south (maritime climate), and the northern and southern, Southern Ocean. Changes in temperature and precipitation for different time slices were presented in Reeves et al. (2013a) relative to the previous time slice for each of these sub-regions. For example, if a sub-region is warmer



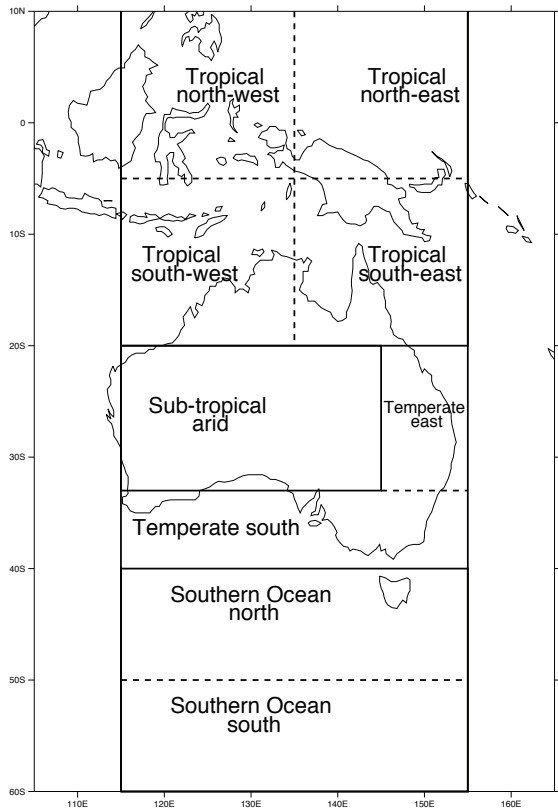

**Figure 1.** A map of the geographical region under consideration in this paper. Overlaid are the borders and names of the regions referred to in the text and correspond to those of Reeves et al. (2013a).

(cooler) in one time slice relative to the previous, the whole box is shaded red (blue). A similar analysis and colour scheme is also applied to effective precipitation. Such an analysis would be easily applicable to surface temperature and precipitation data from GCMs by area averaging over the regions in Fig. 1. The area-averaged temperature and precipitation fields between different periods could then be compared quantitatively from the models and compared directly with the proxy data.

5    Here, we focus is on the mid-Holocene (6000 years before present; 6 ka) experiment of the Paleoclimate Modelling Intercomparison Project (PMIP), in comparison with the pre-industrial era c1750 C.E. (see Section 2 for more details). The OZ-INTIMATE compilations focused on trends from a previous period to the next and generally grouped 6 ka as part of a broader mid-Holocene phase or distinct from the 8 – 7 ka, period with respect to optimal temperature and precipitation conditions (Petherick et al., 2013; Reeves et al., 2013a). In comparison, 6 ka is a fairly "unremarkable" time in the climatic history of
10   Australasia but there is an opportunity to evaluate the impact of the reduced El Niño Southern Oscillation (ENSO) amplitude



(relative to present) and weaker seasonality (lower austral summer insolation) on the climate. Furthermore, this paper presents a case for the development of more seasonally-resolved proxy reconstructions to compare with the models.

The aims of this study are fourfold. Firstly, the study provides a rigorous assessment of the PMIP 6 ka experiments (Braconnot et al., 2007) using the "simplified patterns of temperature and precipitation" method from Reeves et al. (2013a) to show how it can be applied to model data. Secondly, the paper shows where the models and proxies agree or disagree by comparing them directly against each other using the "simplified patterns" method. Thirdly, where the models and proxies agree, then the processes that are responsible for the climatic state are highlighted. Conversely, where the models and proxies conflict to some degree, the (fourth) aim is to provide an explanation as to why the dispute arises. It is not the intention of this work to say the models or the proxy interpretations are incorrect, instead the intent is to show that proxy-model agreement gives confidence in our assessment of past climate and the dynamical mechanisms of the proxy response. Conversely, disagreement provides a *key opportunity* to re-focus our efforts and resolve the issue in an integrated way.

An overview of the data and methods used in this analysis are presented in Section 2. The differences in climatic variables (surface temperature, precipitation and circulation) for 6 ka relative to available reference data are presented for the models and proxies in Section 3. A discussion of where the models and proxies agree and conflict is presented in Section 4 along with an assessment of:

1. What processes are responsible for the climatic conditions at 6 ka when there is agreement and,

2. What may cause any disagreement between the models and proxy reconstructions.

Suggestions of where future efforts should be focused are also presented in Section 4. Finally, a summary of the main results and conclusions are given in Section 5.

## 2 Data, Analysis and external forcings

### 2.1 Proxy data

The OZ-INTIMATE program was developed in 2005 as the Australian component of the international paleoclimate synthesis effort: INTegration of Ice core, MArine and Terrestrial records (INTIMATE), which formed in 1995 as a core program of the INQUA (International Union for Quaternary Research) Palaeoclimate Commission (PalComm).

The OZ-INTIMATE compilation, was built on the four regional reviews (Bostock et al., 2013; Fitzsimmons et al., 2013; Petherick et al., 2013; Reeves et al., 2013b) to develop a climate event stratigraphy for the Australian region (Reeves et al., 2013a). The records chosen for inclusion were representative, rather than comprehensive. The selection criteria for the INTIMATE program are records that are continuous and cover the period of interest, of centennial to millennial scale resolution and have robust chronologies. As this is rarely achieved in Australia, the OZ-INTIMATE synthesis included discontinuous records that are well-dated (given the limitations of the available methodologies and datable material) and include key intervals of change with a reconcilable proxy response to climate were also included. These incorporate a combination of high resolution, centennial or better scale reconstructions (e.g. marine, speleothem, coral records); discontinuous geomorphic records



**Table 1.** The boundary conditions (trace gases) and orbital parameters for the 0 ka and 6 ka PMIP experiments

| Experiment | $CO_2$ (ppmv) | $CH_4$ (ppbv) | $N_2O$ (ppbv) | Obliquity (°) | Eccentricity | Angular (°) |
|---|---|---|---|---|---|---|
| 0 ka | 280 | 760 | 270 | 23.446 | 0.0167724 | 102.04 |
| 6 ka | 280 | 650 | 270 | 24.105 | 0.018682 | 0.87 |

(e.g. fluvial, lake shore, dune, glacier); and well-constrained qualitative and semi-quantitative biological records (pollen, diatom, ostracod, charcoal, geochemistry). Note that records that have been published since 2013 that fit these criteria have been included in our study, where they meet the OZ-INTIMATE criteria and add clarity to previous interpretations—these are cited throughout. Whilst the original interpretations of the records were maintained, the context of the site, limitations of the proxies

and chronological integrity of the records were also considered.

## 2.2 Model simulations and boundary conditions

There is a vast amount of paleoclimate GCM output that is freely available from PMIP (Joussaume and Taylor, 2000; Braconnot et al., 2007, 2012). The PMIP initiative includes data from transient simulations of the last millennium along with time slice simulations of the pre-industrial era c1750 C.E. (0 ka), the mid-Holocene (6 ka) and the Last Glacial Maximum (21 ka). In

this study we make use of coupled atmosphere-ocean general circulation model (AOGCM) simulations run for Phases 2 and 3 of the PMIP (PMIP2 and PMIP3, respectively) for 6 ka. Full details of the experiments run, and evaluations of the simulated responses can be found in Braconnot et al. (2007, 2012); Harrison et al. (2014); Taylor et al. (2012).

Data from the mid-Holocene (6 ka) and the pre-industrial control experiments (0 ka) are used here. The boundary conditions (for example, orbital parameters and greenhouse gas concentrations) are given for the 6 ka and 0 ka simulations in Table 1.

In order to show the impact of the orbital parameter differences, the zonal-mean change in incoming solar radiation at the top of the atmosphere (insolation) is plotted in Fig. 2(a) for 6 ka relative to 0 ka. The 6 ka insolation is lower over much of the Southern Hemisphere (SH) between December to June (minima of < -20 W m$^{-2}$ in December-January) and higher between August and November (maxima of > 40 W m$^{-2}$ in October-November). The zonal-mean difference in insolation over the whole year is plotted in Fig. 2(b). There is lower insolation of 0 W m$^{-2}$ to 1 W m$^{-2}$ between 10°N – 40°S arising from

the increased obliquity at 6 ka (see Table 1). Southward of 50°S, the annual mean insolation is more than 4 W m$^{-2}$ higher southward of 70°S at 6 ka. In Fig. 2(c) and (d), respectively, the insolation is split into two six-month seasonal means, which coincide with the times of year when the highest insolation (October to March—austral summer) and lowest insolation (April to September—austral winter) occurs in the SH. Between October to March (Fig. 2(c)), the zonal mean insolation is 5 – 12 W m$^{-2}$ lower at 6 ka between 10°N – 50°S and higher (0 – 7 W m$^{-2}$) southward of 65°S. Conversely between April and

September the insolation is 5 – 10 W m$^{-2}$ higher at 6 ka northward of approximately 50°S (Fig. 2(d)). Insolation is also higher at 6 ka (approximately 1 – 5 W m$^{-2}$) during April to October southward of 65°S.





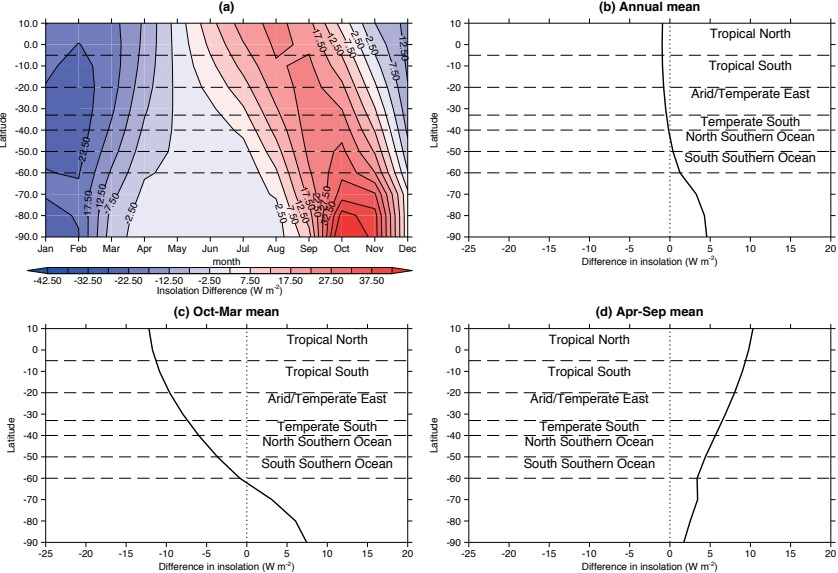

**Figure 2.** The difference in (a) the zonal, seasonal insolation (W m-2) at the top of the atmosphere for 6 ka relative to 0 ka as used by the PMIP2 and PMIP3 models between $10°N - 90°S$. The difference in (b) the annual, (c) October to March and (d) April to September zonal mean insolation (W m-2) at the top of the atmosphere between $10°N - 90°S$.

The only other change applied (under the PMIP framework) to the 6 ka simulations is a reduction in the methane concentrations from 760 ppbv (0 ka) to 650 ppbv (6 ka) (Braconnot et al., 2007). Given that methane concentrations have increased from 760 ppmv (1750 C.E.) to approximately 1800 ppmv (April 2016) and account for approximately 17% of the increased radiative forcing since 1750 (0.5 W m$^{-2}$ out of approximately 3 W m$^{-2}$ total, see Blasing, 2016), the climatological impact of reducing the concentration by 110 ppbv will be negligible.

In all, 32 (18 from PMIP2 and 14 from PMIP3) different model simulations are used in this study, which are listed in Table 2. The original grid configurations of the models (along with the relevant references for each model) are given in Table 2; however, all model data were bilinearly interpolated to a common longitude-latitude grid before undertaking the analysis below for ease of comparison.

## 2.3 Post 1750 C.E. datasets

Two other datasets are used in this study to evaluate the PMIP2 and PMIP3 experiments. Firstly, the Hadley Centre Sea Ice and Sea Surface Temperature data (HadISST, Rayner et al., 2003) from 1870 – 1899 are used to evaluate the simulated sea surface temperature (SST) in the 0 ka and 6 ka experiments over the tropical Pacific Ocean and Southern Ocean. This is done to make use of an instrument-based observational dataset. Secondly, the low-level (850 hPa) zonal flow field from ERA-Interim (Dee et al., 2011) for the period 1979 – 2008 is used to highlight modelled circulation errors over the tropical Pacific Ocean. As





neither of these datasets is representative of the climate at 1750 C.E. (as in the 0 ka simulations), they are only used to highlight known biases in the GCM simulations that may cause discrepancy relative to the proxies.

## 2.4 Analysis

To compare the model simulations with the proxy data, we first calculate the area-weighted average of the climatic variable (in this case temperature or precipitation) for each simulation. An anomaly is calculated by subtracting the value for 0 ka from the value for 6 ka. Two measures of multi-model agreement are then calculated. Firstly, a Student t-test is used to determine whether the multi-model mean is significantly different from zero at the 5% significance level with. Each of the model simulations assumed to be statistically independent of the others. The multi-model ensemble however, comprises multiple different versions of the same modelling frameworks (Table 2 examples include CCSM, CSIRO-Mk3, HadCM/GEM and MRI-CGCM), and so the independence assumption may not be strictly valid. Nevertheless, each different version of the same model uses a different configuration of the parameterized physics (e.g. MRI-CGCM2.3 is configured with and without dynamic vegetation) and could be considered as a different model. The independence assumption therefore, in this situation, provides an unconditional assessment of the models' capabilities for representing the climate at 6 ka that is useful for the comparison with the available proxy data. Secondly, a "model consensus" is derived by calculating the percentage of the models that agree on the sign of the temperature or precipitation anomaly (i.e. positive or negative). A value of 50% implies that 16 models have an increase and 16 models show a decrease in temperature or precipitation at 6 ka relative to 0 ka and therefore there is no clear consensus. If the consensus is above 50% then this indicates that $\geq$17 models agree on an increase or decrease (other examples: 21 models agree = 66%, 25 models agree = 78%, 29 models agree = 91%). The consensus provides a measure of model agreement to quantify how representative the t-test result is across the model ensemble. These two measures of multi-model agreement are used to highlight whether a change in surface temperature or precipitation is a robust feature across the simulations or not. The results of the model analysis are then compared with the available proxy data.

## 3 Model and proxy synthesis

### 3.1 Annual mean

#### 3.1.1 Surface temperature

On average (across the PMIP2 and PMIP3 multi-model ensemble), temperatures in the tropical regions are lower at 6 ka than at 0 ka (Fig. 3(a)). However, the consensus on the sign and magnitude of the temperature changes varies regionally. There is strong agreement across the models (81%) of lower surface temperatures in the tropical north-west (-0.17±0.05 K) and north-east (-0.21±0.05 K) regions when averaged over all simulations. The model agreement is weaker in the southern tropical regions (approximately 60%) as are the lower surface temperatures (-0.09±0.06 K and -0.09±0.05 K for the ensemble mean in the tropical south-west and south-east, respectively). Despite the weaker model agreement in the southern tropical zones, the





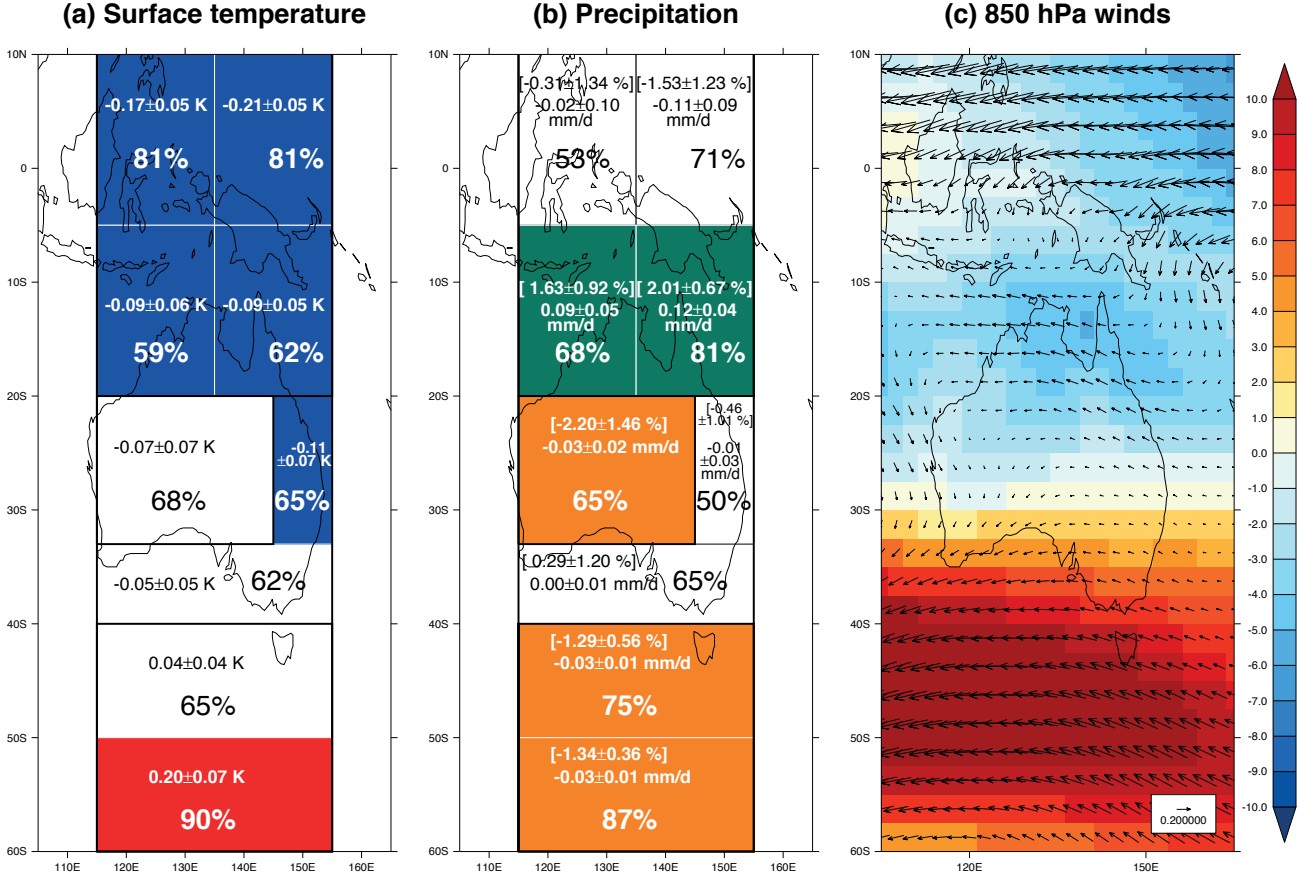

**Figure 3.** The ensemble and regional annual mean differences in (a) surface temperature (K), (b) precipitation (mm day$^{-1}$ and [%]) and (c) 850 hPa circulation (m s$^{-1}$) for the 6 ka simulations relative to the 0 ka simulations. In (a) blue shading represents lower area-averaged surface temperature and red indicates higher at 6 ka. In (b), orange indicates lower area averaged precipitation and green indicates higher at 6 ka. In both (a) and (b) the values of the ensemble mean changes are given in white and the percentages of models that agree on the sign (positive or negative) of the ensemble mean temperature or precipitation differences are given by the white numbers. In (c) shading indicates the direction and strength of the ensemble mean 850 hPa zonal wind (blue colours = easterly and red colours = westerly) in the 0 ka simulations.

lower surface temperature in the 6 ka simulations (relative to 0 ka) is statistically significant (p≤0.05) across the whole tropical region plotted in Fig. 3(a).

Lower temperature at 6 ka relative to 0 ka is also simulated in the sub-tropics and temperate zones (Fig. 3(a)). The lower temperatures at 6 ka are not statistically significant in the sub-tropical arid (-0.07±0.07 K) and southern temperate (-0.05±0.05 K) zones but are statistically significant in the eastern temperate zone (-0.11±0.07 K). Interestingly, despite only one of the





regions displaying a significant change in surface temperature, $\geq$62% of the models agree on lower surface temperatures at 6 ka relative to 0 ka in all three regions (arid, and southern and eastern temperate, Fig. 3(a)). Conversely, surface temperatures were higher on average over both the northern and southern Southern Ocean regions (0.04$\pm$0.04 K and 0.20$\pm$0.07 K, respectively) at 6 ka. The ensemble-mean temperature change is not statistically significant between 40°S – 50°S (despite 65% of the models

agreeing on the sign), but is between 50°S – 60°S (with 90% of the models agreeing).

SST estimates from marine sedimentary records suggest that the Indo Pacific Warm Pool was similar to present during 7-5 ka, with temperatures around 301 K – 302 K (Visser et al., 2003; Stott et al., 2004, 2007; Spooner et al., 2005; Linsley et al., 2010; Leduc et al., 2010). Coral records from the tropical south-west and north-east also support similar SST, relative to present, around 6 ka (Abram et al., 2009).

The tropical south-east proxy data indicates similar to present conditions at 6 ka from marine records (Bostock et al., 2006) and warmer conditions from the terrestrial island records (Woltering et al., 2014) and coral records from the Great Barrier Reef (Gagan et al., 1998, 2004), with slightly lower temperatures in the hinterlands (Haberle, 2005; Burrows et al., 2016). Marginally higher than modern SST are also present along the west coast (Martinez et al., 1999; Spooner et al., 2011), extending through to the Great Australian Bight (temperate southern zone, Calvo et al., 2007; dos Santos et al., 2012). Terrestrial records from

pollen and charcoal from the temperate east and south both indicate slightly lower temperatures at 6 ka than present (Chalson and Martin, 2012; Williams et al., 2015). The Southern Ocean is characterized by higher SSTs both north (Barrows et al., 2007) and south (Crosta et al., 2004) of 40°S, and possible fluctuations in sea ice extent (Ferry et al., 2015) at 6 ka.

### 3.1.2 Precipitation

The largest differences in modelled tropical precipitation at 6 ka relative to 0 ka occur in the eastern zones (Fig. 3(b)) with

lower precipitation (-1.53$\pm$1.23%) in the north-east and higher precipitation (2.01$\pm$0.67%) in the south-east. The difference in precipitation is not statistically significant in the tropical north-east zone. There is high model agreement on the sign of the precipitation change in the tropical south-east (81%) and the simulated higher precipitation at 6 ka is statistically significant. In the western tropical zones, the model consensus is weaker (53% and 68% for the north and south, respectively), as are the relative changes in multi-model mean precipitation (-0.31$\pm$1.34% and 1.63$\pm$0.92%). Nevertheless, the higher 6 ka precipitation

in the south-west tropical zone is statistically significant. In the arid sub-tropical zone, precipitation is (on average) lower in the 6 ka simulations by 2.20$\pm$1.46% (statistically significant) with 65% of the models agreeing.

The eastern and southern temperate zones have very small differences in precipitation for 6 ka relative to 0 ka (-0.46$\pm$1.01% in the east and 0.29$\pm$1.20% in the south) and neither change is statistically significant. However, in the southern temperate zone 65% of the models have higher precipitation in the 6 ka simulation than the 0 ka.

Over the Southern Ocean there is high agreement ($\geq$75%) and statistical significance for reduced precipitation, which is -1.29$\pm$0.60% in the northern zone and -1.34$\pm$0.34% for the southern.

Proxy records from across the region suggest broadly similar precipitation at 6 ka to present (Bostock et al., 2013; Fitzsimmons et al., 2013; Reeves et al., 2013a, b). Slightly drier than modern conditions are recorded in a lake record from Sulawesi (Russell et al., 2014), although marine records from the tropical south-west suggest possible wetter conditions (Stott et al.,





2004) and less variance but similar annual mean precipitation to present in speleothem records from Borneo and Flores (Partin et al., 2007; Griffiths et al., 2009; Chen et al., 2016). The tropical south-east received equivalent or slightly more precipitation at 6 ka than at present in both terrestrial (Kershaw and Nix, 1988; Haberle, 2005; Burrows et al., 2016) and offshore records (Moss and Kershaw, 2007), although drier in the east of the Coral Sea (Duprey et al., 2012) and drier also in the Great Bar-

rier Reef (Lough et al., 2014). Precipitation records from the west are scarce (Denniston et al., 2013), but the arid interior records suggest a drying trend in the monsoon-dominated north and dry conditions, trending to wetter after 6 ka in the westerly influenced south (Fitzsimmons et al., 2013, and references therein).

    Pollen and isotope records from North Stradbroke Island in the temperate east indicate higher precipitation at 6 ka than at present (Moss et al., 2013), although records from Fraser Island suggest lower precipitation (Longmore, 1997; Donders et al.,

2006). The pollen and charcoal records indicate drier conditions in the Sydney Basin and wetter conditions to the south at 6 ka (Chalson and Martin, 2012). Sedimentology, palaeoecology and geochemistry based lake records from western Victoria (southern temperate zone) indicate higher lake levels than present (Kemp et al., 2012); however, in some circumstances lower than their maximum at 7.5 ka (Wilkins et al., 2013). Most records exhibit a long-term trend of lake level decline through 6 ka, which is indicative of relatively dry conditions. A similar pattern is also observed in the transport and deposition (offshore)

of illite clay by the River Murray, indicative of river discharge (Gingele et al., 2007). The pollen and charcoal records from Tasmania reveal overall wetter conditions to the west and drier to the east (Fletcher and Thomas, 2010; Fletcher and Moreno, 2012; Jones et al., accepted; Mariani and Fletcher, in review).

### 3.1.3   Circulation

The ensemble-mean difference (6 ka relative to 0 ka) in the 850 hPa circulation over the region of study is shown in Fig. 3(c).

There are strong easterly anomalies (>0.5 m s$^{-1}$) between 0°N – 10°N and 40°S – 60°S at 6 ka. Over the Southern Ocean, the annual mean wind direction is westerly for the multi-model mean at 0 ka (red shading, Fig. 3(c)) and therefore, given the strength of the easterly anomalies (6 ka relative to 0 ka), the westerlies are weaker at 6 ka (i.e. only easterly anomalies). Over northern Australia, and most of the tropical region shown, the climatological easterlies at 0 ka (blue shading, Fig. 3(c)) are stronger in the 6 ka simulations for the multi-model mean.

Whilst the majority of proxy records do not focus on resolving circulation, in part due to sampling and chronological resolution, the key response of each proxy, the interpretations often invoke changes in circulation as a mechanism to describe alterations to precipitation-evaporation ratios. Records from Queensland in the tropical south-east (Burrows et al., 2016) and the temperate east (Barr et al., 2013) sectors both suggest wetter conditions associated with stronger easterlies. Whilst chronological resolution is generally poor in the arid zone, Quigley et al. (2010) suggest a southerly penetration of the inter-tropical

convergence zone (ITCZ) around 6 ka, evidenced by high lake levels in Frome and Callabonna.

    Records from the western Victorian crater lakes in the temperate south suggest highly variable conditions (i.e. regularly fluctuating between high and low rainfall), with a marked decrease in effective precipitation from 7 ka to 6 ka and in the period prior to 1750 C.E. (e.g. Wilkins et al., 2013; Gouramanis et al., 2013). The primary moisture source for these lakes is the westerlies, therefore they may be assumed to be weakening at this latitude at this time. This is also supported by the records



from the southern, westerly-dominated, arid zone of central Australia (Fitzsimmons et al., 2013). Further south, records from the west of Tasmania, New Zealand and South America show a persistence of wet conditions on the western (windward) flanks of the mountains that intercept westerly flow between 40°S – 44°S across the hemisphere, suggesting that, while possibly attenuating, there was a persistence of relatively strong westerly flow at this latitude at ca. 6 ka (Fletcher and Moreno, 2012).

More recent, high-resolution studies from more ENSO-dominant areas of Tasmania support the 6 ka time-slice as being one marked by a transition to a greater influence of ENSO variability (Mariani and Fletcher, in review; Beck et al., accepted).

## 3.2 Warm season

In this section, the months October to March (inclusive) are considered as the "warm" season (austral summer). This is when insolation in the Southern Hemisphere peaks and is also the time of year when the Australian monsoon rainfall occurs (Sturman

and Tapper, 2006). Conversely, this is also the season that the sub-tropical high-pressure belt is at its most southward extent and therefore the higher latitudes are typically drier (Sturman and Tapper, 2006; Risbey et al., 2009). Therefore, in general, proxies that respond to moisture, such as fluvial and speleothem records, will be more pronounced in the north of the Australasian region during the warm season. Furthermore, as many of the biological proxies are representative of spring or summer conditions, the warm season reconstructions are better constrained than during the autumn or winter.

### 15 3.2.1 Surface temperature

In the north-east tropical zone, the ensemble mean surface temperature is lower at 6 ka relative to 0 ka by -0.10±0.06 K (Fig. 4(a)) and is statistically significant; however, only 59% of the models agree on the sign. In the other tropical regions, the differences between the 6 ka and 0ka ensemble-mean temperature are not significant and the model consensus is <60% in each region (north-west, south-west and south-east).

In both the sub-tropical arid and temperate eastern Australian zones there are lower ensemble mean surface temperatures (-0.18±0.12 K and -0.10±0.09 K, respectively) than in the tropics (Fig. 4(a)) at 6 ka. Furthermore, 78% of the models agree on reduced temperatures over arid Australia; however, 65% agree in the eastern temperate zone and the estimated temperature change is not statistically significant.

The ensemble mean surface temperatures were higher at 6 ka in the southern temperate zone (0.08±0.08 K—not statisti-

cally significant) and both the northern (0.15±0.06 K) and southern (0.26±0.08 K) Southern Ocean zones (Fig. 4(a)). Model agreement also increases southwards with relatively weak model agreement (56%) on temperature differences in the southern temperate zone whereas there is very strong agreement on higher temperatures at 6 ka in both Southern Ocean zones (84% and 90%).

The information we have from proxy records regarding warm season temperatures is largely restricted to the marine realm.

Slightly warmer conditions are evident in all records from the tropical north-east, south-east and south-west zones (Reeves et al., 2013b, and references herein) around 6 ka. Southern Ocean records indicate colder spring, but warmer summer conditions (Sikes et al., 2009).





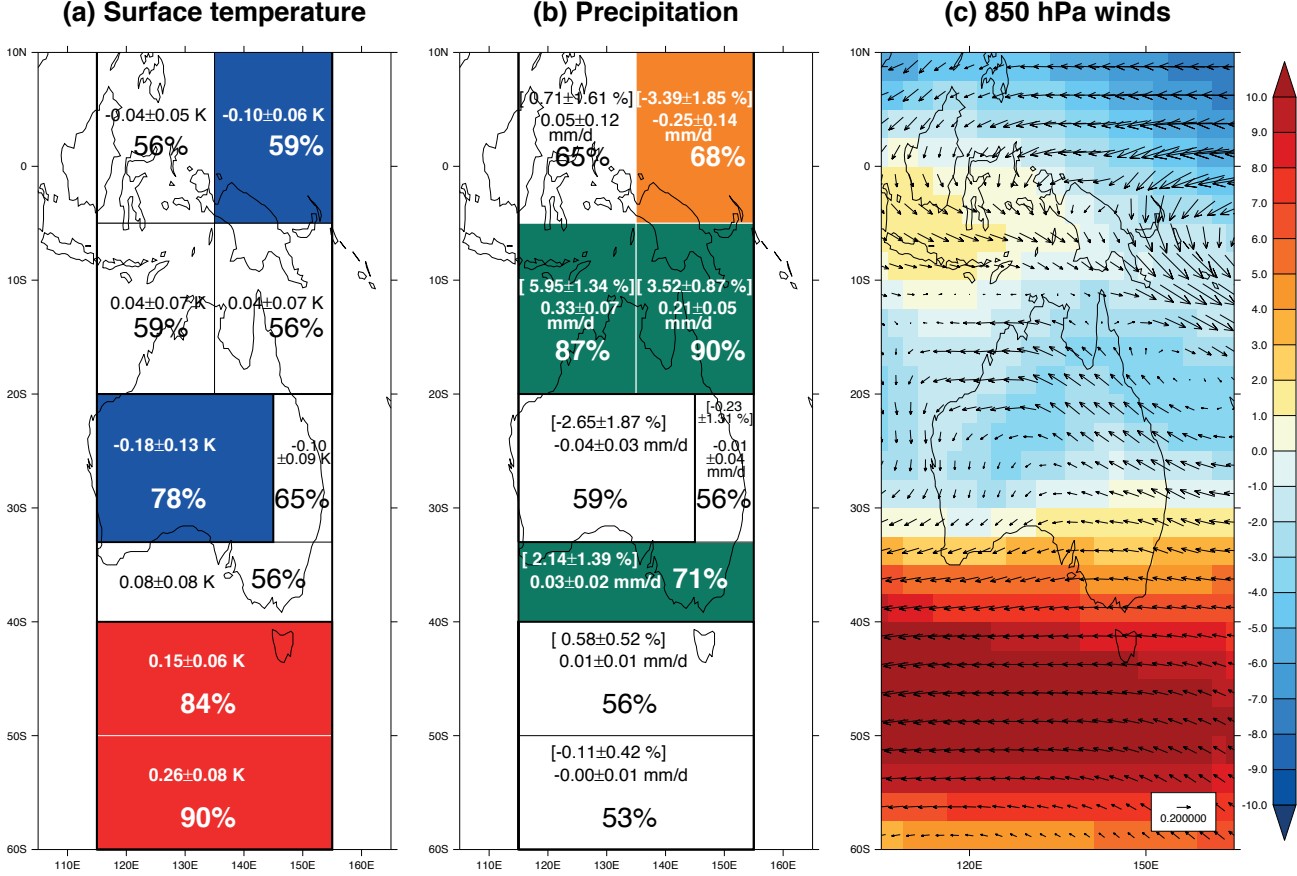

**Figure 4.** The ensemble and regional October to March mean differences in (a) surface temperature (K), (b) precipitation (mm day$^{-1}$ and [%]) and (c) 850 hPa circulation (m s$^{-1}$) for the 6 ka simulations relative to the 0 ka simulations. In (a) blue shading represents lower area-averaged surface temperature and red indicates higher at 6 ka. In (b), orange indicates lower area averaged precipitation and green indicates higher at 6 ka. In both (a) and (b) the values of the ensemble mean changes are given in white and the percentages of models that agree on the sign (positive or negative) of the ensemble mean temperature or precipitation differences are given by the white numbers. In (c) shading indicates the direction and strength of the ensemble mean 850 hPa zonal wind (blue colours = easterly and red colours = westerly) in the 0 ka simulations.

### 3.2.2 Precipitation

In the north-east tropical zone there is lower precipitation at 6 ka (-3.39±1.85%, statistically significant) with 68% of the models agreeing (Fig. 4(b)). There is similar agreement for higher precipitation at 6 ka in the north-west zone (65%); however, the ensemble mean change is only 0.71±1.64% (not significant). Over the tropical southern regions, the 6 ka ensemble mean





precipitation is higher than at 0 ka for both the eastern (3.52±1.02%, significant) and western (5.95±1.27%, significant) zones and also ≥87% of the models agree on the increase at 6 ka relative to 0 ka.

In both the arid and temperate eastern Australian zones, 6 ka ensemble mean precipitation amounts are lower (-2.65±1.87%—significant and -0.23±1.31%—non-significant); however, model agreement on the changes is <60%. Conversely, there is ev-

idence for higher precipitation at 6 ka in the temperate south zone (2.14±1.39%—significant) with >70% model agreement. Finally, there are small differences in precipitation for 6 ka relative to 0 ka over both Southern Ocean zones (ensemble means of 0.58±0.52% and -0.11±0.42%, both non-significant) and model agreement is ≤56%.

Speleothem records from the tropical north-west and south-west show slightly higher warm season precipitation (Denniston et al., 2013) at 6 ka. Pollen records also indicate higher wet-season precipitation in the tropical south-east and the north of

the temperate east regions around 6 ka (Haberle, 2005; Moss and Kershaw, 2007; Moss et al., 2013). Wetter conditions in the northern tropics influenced arid zone of Lake Eyre also implicate a more active monsoon at this time (Magee et al., 2004; Fitzsimmons et al., 2013), although the southern arid zone is drier.

### 3.2.3 Circulation

Ensemble mean easterly anomalies are present in the 6 ka simulations northward of the equator (Fig. 4(c)) and indicate a

strengthening of the easterlies already present in the 0 ka simulations (blue colours). There are westerly anomalies between 0°S – 10°S, which are associated with stronger westerlies in the west of the domain and weakened easterlies in the east at 6 ka. The easterlies over Australia between 20°S – 30°S are stronger at 6 ka. There are also ensemble mean easterly anomalies southward of 30°S; however, the anomalies are associated with a weakening of the westerly flow (indicated by the red colours Fig. 4(c)) at 6 ka.

## 3.3 Cold season

In this section, the months April to September (inclusive) are considered as the "cold" season (austral winter). This is when insolation in the Southern Hemisphere is lowest during the year and also coincides with the dry season in northern Australia. Conversely, this is also the season that the mid-latitude westerly wind belt is at its most equatorward extent and therefore the mid-latitudes are typically wetter (Sturman and Tapper, 2006; Risbey et al., 2009). Therefore, precipitation-driven proxies are

most strongly influenced by cool season anomalies in the higher latitudes. However, as several of the biological proxies are biased toward spring or summer conditions, the cool season reconstructions may be poorly constrained.

### 3.3.1 Surface temperature

There is strong agreement (>80%) for lower surface temperatures in all tropical zones at 6 ka relative to 0 ka (Fig. 5(a)) and the magnitude of the temperature difference is larger in the northern zones (-0.31±0.05 K and -0.32±0.06 K) than the southern

zones (-0.22±0.07 K and -0.23±0.05 K). Furthermore, more than two-thirds of the models indicate lower temperatures at 6 ka in the temperate eastern and southern zones (-0.13±0.07 K and -0.19±0.06 K, both statistically significant), and the northern





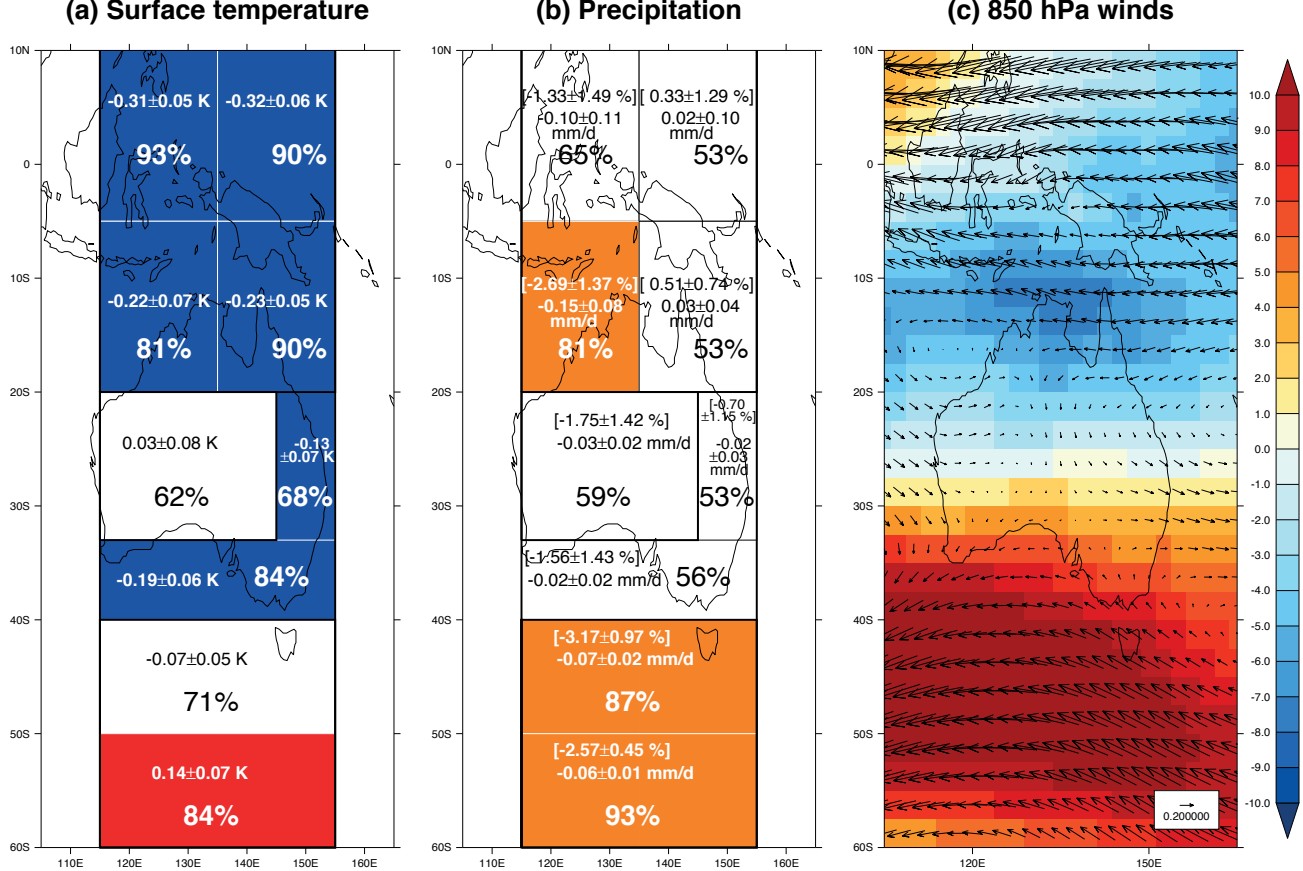

**Figure 5.** The ensemble and regional April to September mean differences in (a) surface temperature (K), (b) precipitation (mm day$^{-1}$ and [%]) and (c) 850 hPa circulation (m s$^{-1}$) for the 6 ka simulations relative to the 0 ka simulations. In (a) blue shading represents lower area-averaged surface temperature and red indicates higher at 6 ka. In (b), orange indicates lower area averaged precipitation and green indicates higher at 6 ka. In both (a) and (b) the values of the ensemble mean changes are given in white and the percentages of models that agree on the sign (positive or negative) of the ensemble mean temperature or precipitation differences are given by the white numbers. In (c) shading indicates the direction and strength of the ensemble mean 850 hPa zonal wind (blue colours = easterly and red colours = westerly) in the 0 ka simulations.

Southern Ocean (-0.07±0.05 K, not statistically significant). The only two regions with higher surface temperatures at 6 ka are the arid continental zone (0.03±0.08 K—non-significant) and the southern, Southern Ocean (0.14±0.07 K—significant). The model agreement on increased temperatures is 84% for the southern, Southern Ocean but is only 62% for the arid continental region.



There is little evidence from the proxy records with regards cool season temperatures. Coral records from the north-east indicate both summer and winter temperatures higher than present (Gagan et al., 2004) and there is evidence for greater fluctuation in the winter sea-ice extent in the Southern Ocean, which may suggest cooler temperatures (Ferry et al., 2015).

### 3.3.2 Precipitation

Ensemble mean precipitation is not significantly different in all tropical regions except the south-west zone (Fig. 5(b)). In the south-west the ensemble mean change in precipitation is -2.69±1.37% (statistically significant with 81% of the models agreeing on the sign). Lower precipitation at 6 ka is also visible in all of the other regions southward of 20°S (Fig. 5(b)) but is only significant over the Southern Ocean. Precipitation is lower in the northern and southern, Southern Ocean zones by -3.17±0.97% and -2.57±0.45%, respectively (both statistically significant). Furthermore, ≥87% of the model simulations produce less precipitation at 6 ka relative to 0 ka in both Southern Ocean regions.

Again, there is little proxy information that resolves cool season precipitation. The southern arid zone does indicate drier winters at 6 ka, leading to sustained aridity (Fitzsimmons et al., 2013). There is also some suggestion of higher precipitation in the Atherton Tableland region of the tropical south-east, with the precipitation of the driest quarter of the year (June to October) being substantially above (double) the modern average of 100 mm between 7.5 – 5 ka (Kershaw and Nix, 1988).

### 3.3.3 Circulation

Between 10°N – 15°S there is a strengthening of the easterlies at 6 ka relative to 0 ka (Fig. 5(c)). The only exception to the strengthened easterlies is in the north-east of the domain where the westerlies have weakened (north-westward of Borneo). There is little change to the 850 hPa circulation between 15°S – 35°S over the Australian continent. Finally, as with the previous time periods described above, there are weaker westerlies over the Southern Ocean at 6 ka relative to 0 ka (ensemble mean).

There is little direct evidence of cool season circulation from the proxy records. There are records of dust flux in the southeastern Australian highlands (Marx et al., 2011), with reduced aeolian activity at 6.5 – 5.5 ka interpreted to indicate northward expansion of the westerlies zone and wetter conditions, followed by increased dust flux and more arid climates 5.5 – 4 ka. However, chronological control for this portion of the record is not optimal (Marx et al., 2011).

## 4 Mechanisms responsible for agreement and disagreement

### 4.1 Model-proxy agreement

This section highlights areas where there is agreement between the model and proxy estimates (all regions except the tropical north-east and southern temperate) of surface temperature, precipitation and circulation and identifies the physical processes that were responsible for those conditions at 6 ka.





### 4.1.1 Tropical north-west (TNW)

There is evidence of slightly lower (∼0.2 K) SST for 6 ka relative to 0 ka from the model dataset that is consistent with the "similar to or slightly lower SST" estimate from the proxies (Section 3.2) over the TNW. The lower SST in both datasets is consistent with the lower annual mean insolation across the northern tropics (see Fig. 2).

As discussed, the proxy data suggest that there is evidence of slightly lower precipitation within the TNW region at 6 ka. The ensemble mean change in precipitation over the whole TNW domain is -0.3%, but is not statistically significant (Fig. 3(b)). The proxy and model results are still consistent and indicate that differences in overall precipitation in the TNW zone are likely to have been small at 6 ka relative to 0 ka. On a seasonal timescale, both the models and proxies suggest higher October to March precipitation, but the modeled change is not statistically significant—although 65% of the models simulate higher rainfall at

6 ka. Therefore, given that two-thirds of the models have higher warm season precipitation at 6 ka, it is worth taking a closer look at the mean seasonal cycle (from the GCMs) to identify whether the lack of statistical significance is due to the choice of October to March averaging period.

 Initially, an evaluation of modeled seasonal cycle relative to the real world is necessary to confirm whether the GCMs are representing the correct processes. Precipitation is perennially high over the TNW region (McBride, 1998) ; however, there is

a distinct seasonal cycle with higher rainfall from November to April (Lee and Wang, 2014). The rainfall seasonality is not primarily driven by insolation but by the seasonal cycle in the large-scale circulation from a relatively dry southeasterly flow in April to October to a relatively moist northeasterly flow in November to March (Chang et al., 2016; Vincent, 1998). In both the 6 ka and 0 ka simulations, the peaks in insolation (Fig. 6(a)) and surface temperature (Fig. 6(b)) occur in September/October and March-April-May. The highest precipitation occurs in November to April (in the 0 ka and 6 ka simulations, Fig. 6(c)) for both

total (blue line) and convective (turquoise lines) precipitation. The models also represent the seasonal change in wind direction from southeasterly during April to October to northeasterly during November to March (not shown), which corresponds with the seasonal peak in rainfall. Therefore, the seasonal cycle of precipitation is circulation-driven and is not directly related to the seasonal cycle of insolation or surface temperatures in agreement with Chang et al. (2016); Vincent (1998).

 At 6 ka (relative to 0 ka), insolation and surface temperature are higher in August to November, but precipitation is higher

in November to March (Fig. 6(d)), which indicates that the precipitation change at 6 ka is not thermally driven (i.e. directly from higher insolation and surface temperatures). The higher November to March precipitation at 6 ka actually corresponds with anomalous northeasterly 850 hPa flow over the TNW during October to March (Fig. 4(c)), which again shows that the seasonal cycle is driven by the large-scale circulation. Interestingly, the precipitation is lower only for October at 6 ka despite higher precipitation in November to March. It is therefore likely that the non-significant October to March mean precipitation

difference (6 ka relative to 0 ka) presented in Fig. 4(b) is primarily due to the lower 6 ka October rainfall only. Therefore, if October is removed from the model analysis, the agreement between the models and the proxies becomes stronger. Such a sub-seasonal consideration is important in order to address small disparities between models and proxies in order to resolve them.





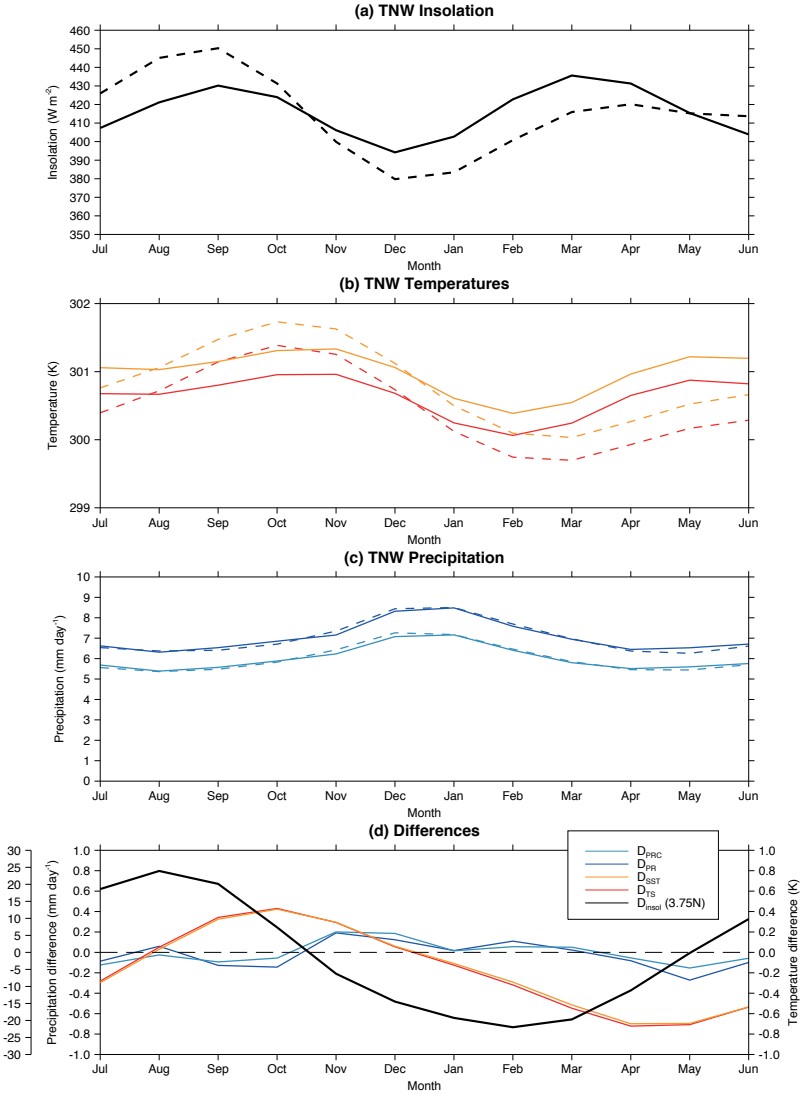

**Figure 6.** The monthly, ensemble and regional mean (a) insolation (taken at 3.75°N for insolation, black line, W m$^{-2}$), (b) surface temperature (land and ocean combined, red line, K) and sea surface temperature (when available, amber line, K) and, (c) total precipitation (blue line, mm day$^{-1}$) and convective precipitation (turquoise line, mm day$^{-1}$) at 0 ka (solid lines) and 6 ka (dashed lines) within the Tropical north-west box. The difference in those fields (insolation, temperature and precipitation) for 6 ka minus 0 ka is plotted in (d).





### 4.1.2 Southern Tropics

The proxies suggest that annual mean precipitation at 6 ka was likely to be similar to the present day or slightly higher in the tropical south-east (TSE). By comparison, 81% of the models simulate higher precipitation at 6 ka relative to 0 ka (Fig. 3(b)) and 90% have higher rainfall in October to March (for 6 ka relative to 0 ka, Fig. 4(b)). The proxy records indicate the higher

precipitation results from stronger easterlies, which is also consistent with the circulation change in Fig. 3(c). Furthermore, the multi-model mean difference in surface temperature between the 6 ka and 0 ka simulations is less than -0.1 K and only 62% of the models agree on lower temperatures, which suggests that the models are fairly evenly distributed around zero temperature change. This is also consistent with the proxy evidence given in Section 3.1.1.

The cause of the higher rainfall at 6 ka (particularly in October to March) over tropical south-east can be seen when the

seasonal cycle of precipitation is considered (Fig. 7). The higher insolation in June to December (Figs. 7(a) and (d)) causes SST at 6 ka to be higher in August to January (Figs. 7(b) and (d), SST response lags the insolation change), which coincides with the period where both the convective and total precipitation are higher in the 6 ka simulations relative to 0 ka (Fig. 7(c) and (d)). Given that onshore flow and sea breeze convergence are important processes that govern precipitation over Cape York Peninsula (i.e. north-east Australia, see Birch et al., 2015), the higher continental temperature (September to December,

Fig. 7(b)) are likely to have enhanced onshore flow. Higher SST adjacent to the coastline would also increase the moisture content of air transported over the land. The earlier onset of the monsoon from increased continental heating is therefore likely to be responsible for the increase in precipitation over the tropical south-east. Conversely, the impact of reduced land and sea temperatures from April to July has little impact on precipitation during the dry season.

Within the tropical south-west (TSW) domain, slightly lower surface temperatures may have prevailed around 6 ka relative

to the preindustrial according to the proxies (Section 3.1.1), which is consistent with the lower annual mean insolation at 6 ka (see Fig. 2(b)). The models also simulate lower mean annual surface temperature at 6 ka relative to 0 ka on average, (approximately -0.1 K) but <60% of the models agree on the reduced temperature. The ensemble-mean change in temperature is however, statistically significant and consistent with the proxy evidence. It is therefore likely that the lower insolation at 6 ka was responsible for the lower temperatures.

The proxy data indicate that rainfall at 6 ka may have also been similar to, or slightly higher than, present over TSW. There is also proxy evidence for higher rainfall in the late dry season, which may be indicative of a shift to an earlier monsoon onset. Indeed, 68% of the models agree on an annual mean increase in precipitation over tropical south-west at 6 ka (Fig. 3(b)) with 87% agreeing on higher October to March precipitation (Fig. 4(b)). Conversely, 81% of the models suggest precipitation was lower in April to September at 6 ka relative to 0 ka (Fig. 5(b)). When the tropical south-west mean insolation and surface

temperatures are plotted seasonally (Fig. 8(a), (b) and (d), it can be seen that higher insolation (and surface temperatures) during October to December corresponds with higher rainfall around the same time (Fig. 8(c) and (d)). Furthermore, the higher simulated 6 ka rainfall is primarily from convection (turquoise line, Fig. 8(c) and (d)) indicating a thermally-driven, direct response to the change in seasonal insolation at 6 ka relative to 0 ka. The models therefore agree with the proxies for higher





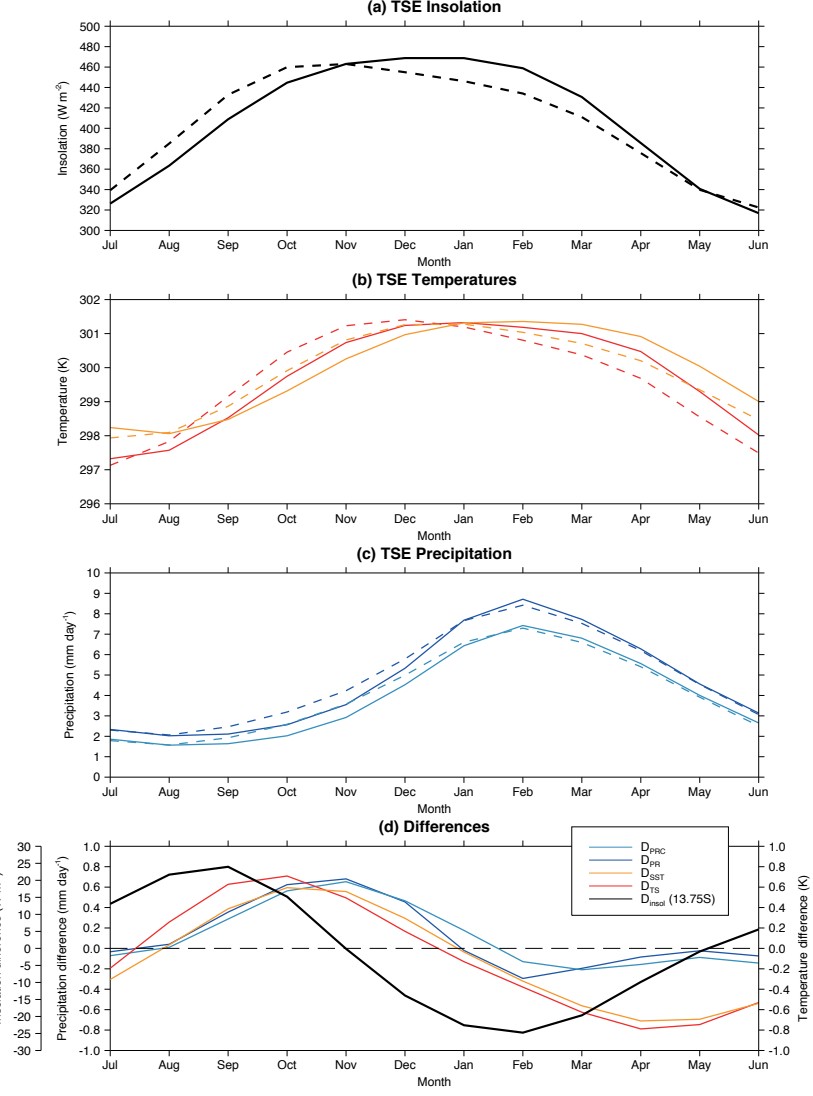

**Figure 7.** The monthly, ensemble and regional mean (a) insolation (taken at 13.75°S for insolation, black line, W m$^{-2}$), (b) surface temperature (land and ocean combined, red line, K) and sea surface temperature (when available, amber line, K) and, (c) total precipitation (blue line, mm day$^{-1}$) and convective precipitation (turquoise line, mm day$^{-1}$) at 0 ka (solid lines) and 6 ka (dashed lines) within the Tropical south-east box. The difference in those fields (insolation, temperature and precipitation) for 6 ka minus 0 ka is plotted in (d).



rainfall at the end of the dry season. Furthermore, the model simulations suggest that the higher 6 ka precipitation is insolation driven.

The temperature and precipitation characteristics of both of the tropical south (east and west) domains appear to respond directly to insolation (Figs. 7 and 8, respectively). In both regions, lower annual mean insolation causes surface temperatures to

be lower around 6 ka relative to 0 ka; however, the lower annual mean temperatures do not result in reduced precipitation. The higher insolation from July to November at 6 ka relative to 0 ka causes the wet season precipitation to start earlier (September – October) than at 0 ka (October – November). As the response of the SST lags the insolation changes by 1-2 months, the average difference in SST in the 0 ka and 6 ka simulations during the middle of the wet season (December to February) are approximately ±0.2 K (i.e. very little difference). Therefore, given the insolation and resulting SST conditions, an overall

increase in wet season precipitation occurs.

### 4.1.3   The arid zone

The sub-tropical arid zone incorporates much of the Australian continent and is sensitive to both the strength of the monsoon in the north and the mid-latitude westerlies in the south. The evidence from the proxy data indicates that there was lower overall precipitation in the monsoon-dominated north half of the zone, or at least suppressed penetration of the ITCZ, with the opposite

true in the southern half (precipitation increasing) at 6 ka. Given the models only simulate conditions at each time slice (i.e. 0 ka and 6 ka) it is difficult to reconcile them with the proxies. Nevertheless, by looking at the seasonal characteristics of the precipitation over the northern and southern halves of the domain, as well as the arid zone as a whole, there may be evidence in the simulations to corroborate the proxy synopsis.

For the overall sub-tropical arid region, the models simulate slightly lower precipitation over the arid zone as a whole for

6 ka relative to 0 ka; however, for the northern half of the arid zone (north of 26.5°S) precipitation is 3.91% lower, and only 0.4% lower for the southern half (south of 26.5°S). Therefore, the change in annual mean precipitation is primarily through a reduction in the monsoon-dominated northern half. Moreover, October to March precipitation in the northern arid zone is lower by 3.57% whereas the change is approximately zero in the southern half for the same months. Interestingly, during April to September, when the southern half of the arid zone is influenced by mid-latitude systems, precipitation is 2.35% lower in

the 6 ka simulations than in the 0 ka simulations.

The seasonal cycles of insolation, surface temperature, convective precipitation and all precipitation are plotted for the northern arid zone in Fig. 9(a). Insolation peaks in December at both 6 ka and 0 ka; however, surface temperatures peak in December at 6 ka and January for 0 ka. Precipitation is higher in July to December when insolation and / or surface temperatures are higher, but there is lower rainfall from January to June when the surface temperatures and / or insolation are lower. The

precipitation appears to be responding more directly to the land surface (and insolation) than in the tropical south-east and south-west domains. Given there is an overall reduction in insolation over the northern arid zone, both the overall mean and convective precipitation are lower at 6 ka despite higher rainfall in July to December (Fig. 9(b)). Moreover, such a direct response of the precipitation to the insolation supports the idea of the monsoon penetrating less southward around 6 ka.



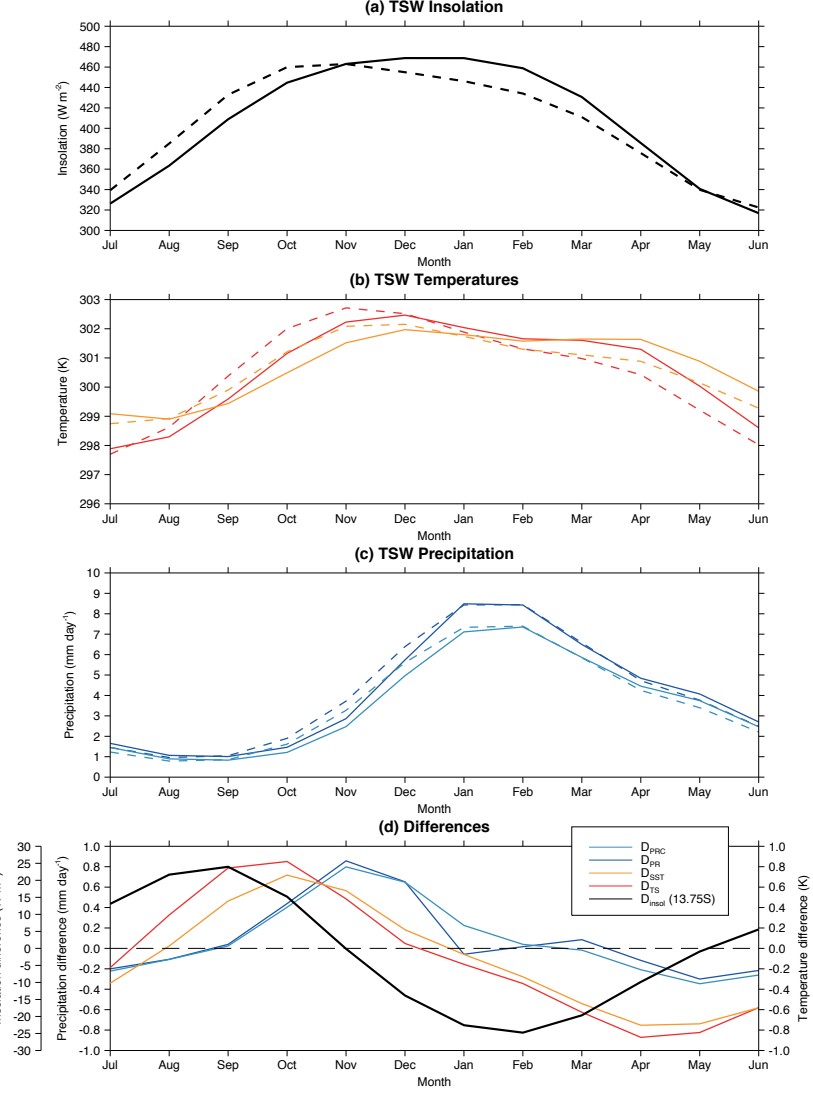

**Figure 8.** The monthly, ensemble and regional mean (a) insolation (taken at 13.75°S for insolation, black line, W m$^{-2}$), (b) surface temperature (land and ocean combined, red line, K) and sea surface temperature (when available, amber line, K) and, (c) total precipitation (blue line, mm day$^{-1}$) and convective precipitation (turquoise line, mm day$^{-1}$) at 0 ka (solid lines) and 6 ka (dashed lines) within the Tropical south-west box. The difference in those fields (insolation, temperature and precipitation) for 6 ka minus 0 ka is plotted in (d).





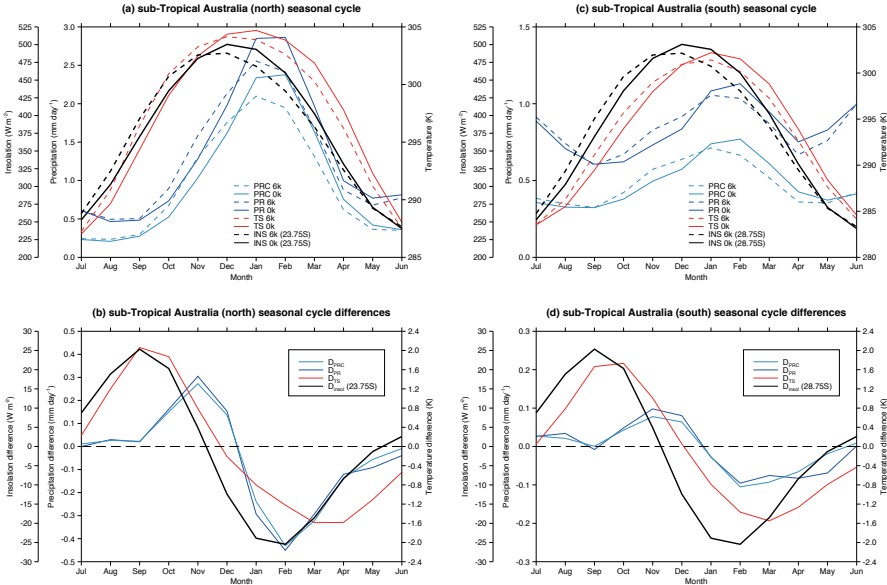

**Figure 9.** The monthly, ensemble and regional mean insolation (taken at 23.75°S for insolation, black line, W m$^{-2}$), surface temperature (red line, K), total precipitation (blue line, mm day$^{-1}$) and convective precipitation (turquoise line, mm day$^{-1}$) at 0 ka (solid lines) and 6 ka (dashed lines) within (a) the northern half of the sub-Tropical arid box and (b) the difference in those same fields for 6 ka minus 0 ka for the northern half of the sub-Tropical arid zone. Equivalent figures for the southern half of the sub-Tropical arid zone (c and d, insolation at 28.75°S) are also plotted.

In the southern half of the arid zone, a similar thermally direct response to the insolation also appears in December to February (Fig. 9(c) and (d)). Nevertheless, there is also a second peak in rainfall in June and July, which is likely to be associated with extratropical systems (Catto et al., 2012). There is lower January to April convective precipitation at 6 ka (Fig. 9(d)), which is consistent with the reduced insolation and surface temperature. Conversely, the increase in insolation and
5  surface temperature causes higher convective precipitation in October to December at 6 ka. Interestingly, there is an increase in precipitation (albeit weak) in July to September, which may be indicative of an increasing influence of extratropical weather systems during the winter to early spring; however, the increase in precipitation appears to be from increased convection (turquoise line, Fig. 9(d)) and indicates that the higher rainfall may be a thermally direct response to the increased insolation in July to September. Regardless of whether convection or synoptic-scale systems are responsible for causing higher precipitation
10  at 6 ka relative to 0 ka in the models, there is evidence of increased mid-to-late winter precipitation that is also detected in the proxy records, outlined earlier.

### 4.1.4 Temperate east

The temperate eastern zone extends from the southern boundary of the tropical zone to the northern boundary of the southern temperate zone but is climatologically distinct from the sub-tropical arid interior as precipitation is higher and varies less



interannually. The evidence from the proxy data indicates lower temperatures at 6 ka relative to 0 ka and a tendency for more easterly flow (see Section 3.1). The proxy evidence is consistent with the model results (Figs. 3(a) and (c)), which show lower surface temperature (-0.11 K) and anomalous easterly flow. Given the lower annual mean insolation between 20°S to 33°S at 6 ka (Fig. 2(b)), it is likely that the models and proxies are responding primarily to the change.

Some uncertainty in the temperate eastern zone results from disagreements between precipitation records from North Stradbroke Island, which indicates higher precipitation at 6 ka, compared to a record from Fraser Island, which indicates lower rainfall (see Section 3.1.2). Nevertheless, given that the models indicate there was no change in precipitation (Fig. 3(b)), it is plausible that the records may be indicative of local climatic effects or subtle shifts in boundary conditions. If more proxy data were available over a larger area then the overall picture may indicate no change. Given the agreement between the proxies and

models for surface temperature and circulation, it seems likely that over the broad area, precipitation at 6 ka may have been similar to present.

### 4.1.5   Southern Ocean

The proxies suggest that SSTs were around 284.2 K in the northern Southern Ocean (NSO) region and approximately 278.7 K in the southern, Southern Ocean (SSO) zone for the annual mean at 6 ka. The HadISST 1870 – 1899 temperatures for the same

regions are 284.0 K (NSO) and 277.0 K (SSO). Therefore, SSTs in the NSO are almost identical at 6 ka to 0 ka whereas in the SSO (50°S – 60°S) the SSTs may have been considerably higher at 6 ka.

Boxplots of the area mean SST for the 0 ka (white boxes, left axis) and 6 ka (amber boxes, left axis) simulations are plotted in Fig. 10 in order to compare the HadISST (red circles) and proxy (blue squares) data to the models. The horizontal line within the box denotes the median, which is used to denote the middle value here instead of the mean; however, there is little

difference between the median and mean in the results presented so the median is used here to comply with the format of the boxplots.

For the NSO, the model median SST is within 0.1 K of the HadISST estimate (284.0 K and 283.9 K, respectively) with 50% of the models simulating SSTs within approximately 1 K of the HadISST estimate (lower quartile model average SST is 282.7 K and the upper quartile model average is 284.7 K). For the 6 ka simulation, there is no change in the median SST (283.9 K)

and little change in the interquartile range (282.8 K to 284.7 K). The difference in the multi-model median SST between the 6 ka and 0 ka simulations (6 ka minus 0 ka for each model individually) is almost zero (0.03 K, see pink boxes in Fig. 10, right axis). Therefore, the models and proxies agree that the SSTs in the NSO region at 6 ka are likely to have been very similar to 0 ka. The lack of any SST change in NSO is also consistent with the insolation changes between 40°S – 50°S, which are negligible (see Fig. 2(b)).

Boxplots for the area mean SST for the 0 ka (white boxes) and 6 ka (amber boxes) simulations are also plotted for the SSO region in Fig. 10 along with the differences (6 ka ? 0 ka) for each individual model (pink boxes). The model median SST is 0.5 K higher than the HadISST estimate (277.5 K compared to 277.0 K, respectively), but lies within the simulated interquartile range (276.6 K to 278.3 K). In the 6 ka simulations, the median model mean SST is 277.7 K and the upper and lower quartiles are 277.0 K and 279.0 K, respectively and are therefore higher than in the 0 ka simulations. This can also be seen for the 6 ka



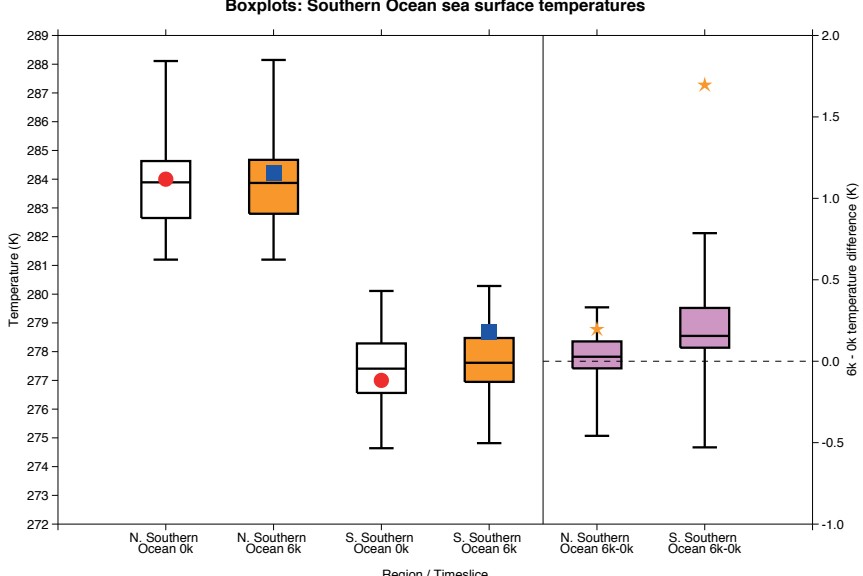

**Figure 10.** Box-and-whisker plots of the absolute mean temperature (K) in the north, Southern Ocean (NSO) box at 0 ka (first white box on left) and 6 ka (first amber box on the left), and for the south, Southern Ocean (SSO) at 0 ka (second white box from the left) and 6 ka (second amber box from the left). The difference between the 6 ka and 0 ka NSO and SSO model mean temperatures (K) are plotted in the leftmost and rightmost pink boxes, respectively. The red circle indicates the value from HadISST, the blue square is the proxy reconstruction estimate and the amber stars indicate the difference between the blue squares and red circle (i.e. past minus reference period).

– 0 ka boxes (pink) where the median simulated change in SST is approximately 0.16 K. Nonetheless, the models typically underestimate the change in SST relative to the proxy estimate, which lies close to the upper quartile model and outside the range of the SST differences for 6 ka relative to 0 ka (amber star above pink box, Fig. 10).

There are acknowledged spatial and temporal gaps in the proxy data for the Southern Ocean (Bostock et al., 2013); however,
5    there are also significant known deficiencies in the model simulations within this region. While it is commendable that the models represent the sign of the SST change in both of the Southern Ocean regions at 6 ka (NSO and SSO), the small SST response to the higher insolation at 6 ka may be the result of several known errors that lie within the models used in PMIP2 and PMIP3 (from the CMIP3 and CMIP5 generation of models, respectively). These errors are discussed here to account for the weak response of the models to the increased insolation, despite the model-proxy agreement in the sign of the temperature
10    change.

Trenberth and Fasullo (2010) have shown that the cloud cover fraction over the Southern Ocean is too low within the CMIP3 models, which leads to a positive bias in the amount of solar radiation absorbed at the ocean surface. Furthermore, this cloud-related bias in the absorbed solar radiation (approximately +8 W m$^{-2}$, Schneider and Reusch, 2016; Flato et al., 2013) is still present in the next generation of coupled climate models (CMIP5), which corresponds with positive SST biases (see Figs. 9.2





and 9.5 in Flato et al., 2013). Approximately 70% of the models simulate higher SSTs between 50°S – 60°S at 0 ka relative to HadISST (Fig. 10), which is consistent with the known cloud cover and radiation errors described above.

It is not immediately obvious how a negative bias in cloud cover fraction (and positive bias in absorbed solar radiation) could dampen the expected SST response in the 6 ka simulations relative to the 0 ka simulations; however, the Antarctic Circumpolar Current (ACC) runs through the SSO zone in these models (Gupta et al., 2009; Meijers et al., 2012) and both the NSO and SSO regions lie within the zone where the deepest winter mixed layer depths form in the Southern Ocean (Meijers, 2014; Sallée et al., 2013a). The winter mixed layer depths in the models are generally too shallow relative to the observations (Meijers, 2014; Sallée et al., 2013a), which is associated with a positive temperature bias in the water formed in this region (Sallée et al., 2013b). Furthermore, Sallée et al. (2013a) also show that simulated mixed layer depths become shallower (shoaling) under a global warming scenario; however, the shoaling of the mixed layer was relatively larger in those models with a deeper mixed layer in the historical simulations, which indicates that the global warming-induced response is dependent on the initial state. Such a process may be responsible for the muted response of the SSTs to the increase in insolation (i.e. a poor representation of the ocean processes weakens the forcing response).

Other work by Wang et al. (2014) shows that the positive SST biases are strongest in the Southern Hemisphere summer and autumn, which corresponds with the time that the cloud cover related errors in the absorbed solar radiation are at a maximum (i.e. when insolation is highest). Nevertheless, Wang et al. (2014) attribute the Southern Ocean warm bias to errors to the simulation of the meridional overturning circulation and not the cloud radiative errors. Overall, it is clear that there are large errors in the Southern Ocean circulation within GCMs that could be caused by several different processes (e.g. cloud radiative forcing, ocean mixed layer depth and the meridional overturning circulation), which may dampen or enhance the SST response to a change in insolation. Therefore, the small change in simulated SST in the Southern Ocean (<0.3 K) compared to the proxy data (>1 K) may be a manifestation of these combined errors and provides further evidence that the representation of the atmosphere and ocean at high southern latitudes should be a priority for the modelling community.

## 4.2 Model-proxy conflict

The two regions in which the modelling and proxy records do not agree (tropical north-east and the temperate south) are discussed here.

### 4.2.1 Tropical north-east (TNE)

The tropical north-east region includes the eastern side of the Western Tropical Pacific Warm Pool and is therefore an important indicator for the ENSO. Proxy data indicate that SSTs were higher around 6 ka relative to present during a period of reduced ENSO variance between 7 – 5 ka (Tudhope et al., 2001; McGregor and Gagan, 2004; Abram et al., 2009; Emile-Geay et al., 2016). Conversely, 81% of the models analysed simulate a reduction in surface temperature within the tropical north-east zone (-0.21±0.05 K). Therefore, there is a strong disagreement between the proxy data and the models and, given the high confidence in the proxy records, it is likely to be the models that are incorrect.



Many coupled, ocean-atmosphere GCMs are known to represent the SST field across the equatorial Tropical Pacific poorly. Typically, the SSTs are too low along the equatorial Pacific and those negative SST errors extend into the western Tropical Pacific (Brown et al., 2013; Grose et al., 2014; Irving et al., 2011; Zheng et al., 2012), which is known as the "cold-tongue bias". Furthermore, the same errors are also visible in the PMIP2 and PMIP3 simulations used in this study (An and Choi,

2014). A simple way to remove the impact of the error is to assume that it remains unchanged in a different climatic state (such as changing the Earth's orbital parameters). Such an assumption implies that the difference between two simulated climate states is representative of the "observed" (e.g. proxy data) difference despite the initial error. Nevertheless, it seems that for 6 ka conditions, the SST errors may actually be enhanced and therefore the change in the climate state is dependent on the initial error in the background state.

Previous work by Clement et al. (2000) suggest that there was a strengthening of the Pacific trade winds in the austral spring around 6 ka, which has been attributed to a strengthening of the south-east Asian summer monsoon and is also represented by the PMIP2 and PMIP3 models (An and Choi, 2014). Nevertheless, if the Tropical Pacific easterlies are already too strong in the 0 ka simulations, any further strengthening could enhance the existing cold tongue bias through the Bjerknes feedback mechanism (Bjerknes, 1969; Lin, 2007). To illustrate this, the difference between the PMIP ensemble mean and HadISST

(1870 – 1899) SSTs are plotted in Fig. 11(a). Overlaid on the figure are the differences in the 850 hPa zonal wind speed for the ensemble mean relative to ERA-Interim (averaged over 1979 – 2008). It is immediately obvious that there is a strong (<-1.5 K) SST anomaly along the western equatorial Pacific, which coincides with an easterly zonal wind bias. Recent work by Li and Xie (2014) suggests that the zonal wind error is responsible for the cold tongue bias through the Bjerknes feedback; however, Zheng et al. (2012) suggest other mechanisms may be responsible. Regardless of the actual cause, both the cold tongue bias

and easterly errors are present in the PMIP ensemble in Fig. 11(a).

Both Zhao and Harrison (2012) and An and Choi (2014) show that the easterly flow is enhanced in the PMIP2 and PMIP3 simulations of 6 ka relative to 0 ka, which is attributed to an enhancement of the south-east Asian monsoon. Both the easterlies and negative SST anomalies (relative to HadISST and ERA-Interim) are stronger in the 6 ka simulations relative to the 0 ka simulations (PMIP2/3 multi-model mean, Fig. 11(b)). Therefore, the difference between the 6 ka and 0 ka simulations (Fig.

11(c)) is primarily from the enhancement of the errors that already exist in the 0 ka simulations. In reality, stronger easterly flow from an enhanced south-east Asian monsoon should act to deepen the thermocline in the western Pacific warm pool and increase SST (in agreement with aforementioned proxy data). Nevertheless, given the large error in the model-simulated initial state, stronger easterlies in the 6 ka simulations actually cause the modeled SSTs to reduce in the western Pacific. Given that these model errors are exacerbated under a different set of orbital parameters, it may imply that the error may also be

sensitive to other boundary condition perturbations (e.g. increasing atmospheric $CO_2$). Overall, this result indicates that fixing the cold-tongue bias is still a very high priority for the climate modelling community.

### 4.2.2 Temperate south

The proxy records indicate higher SSTs in the Great Australian Bight and slightly lower terrestrial temperatures within the temperate south zone (Calvo et al., 2007; dos Santos et al., 2012) (see Section 3) at 6 ka. The models simulate lower (but





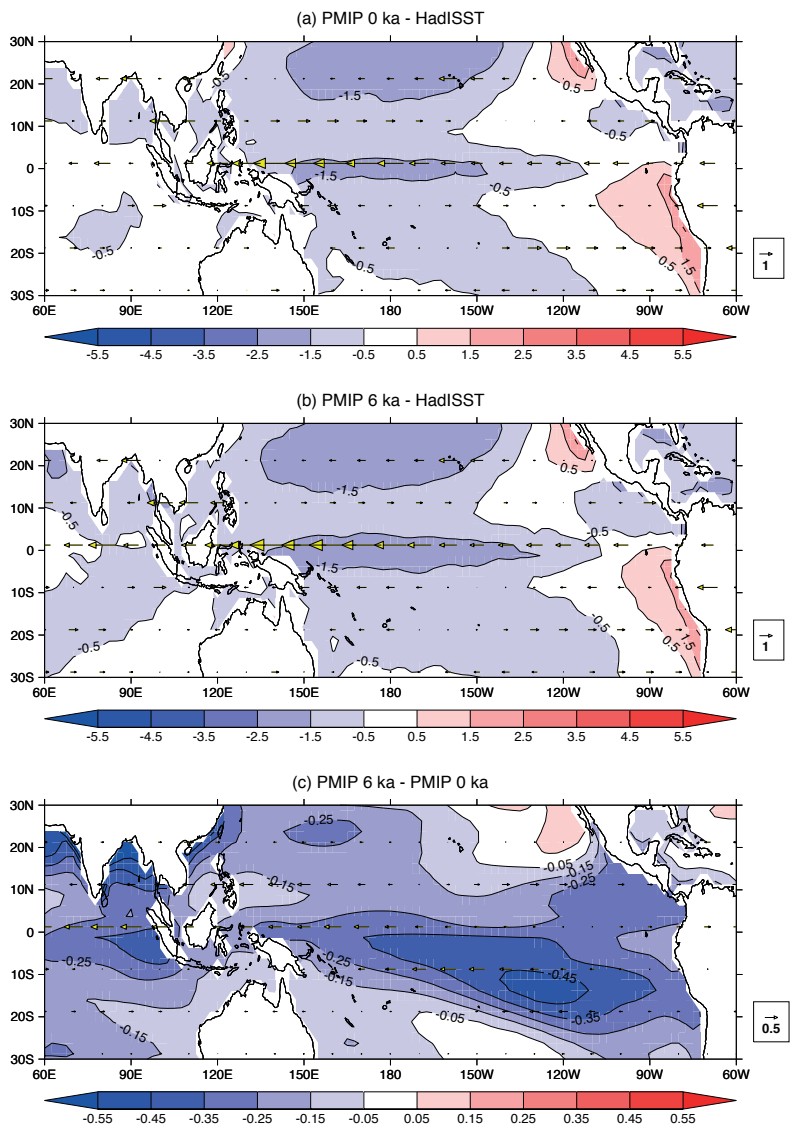

**Figure 11.** The multi-model mean difference in SST (shading, K) and 850 hPa flow (arrows, m s$^{-1}$) for the 0 ka simulations relative to HadISST (1870 – 1899 average) and ERA-Interim (1979 – 2008 average), respectively. (b) the same as (a) except for the 6 ka simulations. (c) The multi-model mean difference in SST and 850 hPa flow for the 6 ka simulations relative to the 0 ka simulations.



non-statistically significant) area averaged and ensemble mean surface temperatures at 6 ka relative to 0 ka (-0.05 K, Fig. 3(a)), which is consistent with the terrestrial record, but not the SST record (Chalson and Martin, 2012; Williams et al., 2015). Given the complex ocean circulation within the Bight and the coarse resolution of the models (see Table 2), it is not surprising that there is disagreement between the models and the proxies (i.e. the models cannot resolve the processes that the proxies can).

Accepting the reduction in insolation and the direct thermal response of the land (which has been shown to be represented well in general in Section 4.1), the reduction in terrestrial temperature is represented (albeit non-statistically significant).

The first point of difference between the model simulations and the proxies regard the changes in circulation (mid-latitude westerlies) and precipitation. The proxy data indicate that slightly wetter conditions prevailed (at 6 ka) in the annual mean for the southern temperate zone, which is attributed to either a weakened sub-tropical ridge or stronger southwesterly winds.

Similarly, the models simulations indicate no change / slightly higher precipitation (only +0.3%) at 6 ka relative to 0 ka with 65% of models having higher precipitation. Accepting that a reduction in the strength of the mean westerly flow over southern Australia (Fig. 3(c)), and assuming that contemporary correlations between present day precipitation and the strength of the westerlies apply in the past, it might be expected that precipitation should reduce significantly in the models. There is clearly no evidence of lower precipitation at 6 ka despite the weaker westerlies. Furthermore, 71% of the models actually simulate

more precipitation in October to March (+2%, Fig. 4(b)) when the sub-tropical high-pressure cell dominates the circulation (Sturman and Tapper, 2006). Interestingly, there is little consensus on reduced April to September rainfall (56% of the models simulate reduced precipitation) when the westerlies have their greatest influence on southern Australia precipitation through frontal cyclones (Catto et al., 2012).

The strength of the westerly winds and their influence on precipitation in this region around 6 ka has been widely discussed

in the literature (e.g. Fletcher and Moreno, 2012; Gouramanis et al., 2013); however, it is clear from the models that the assumption that stronger (weaker) westerlies lead to higher (lower) mean precipitation may not always be true. As the physical processes responsible for precipitation can be diagnosed directly from the PMIP simulations, there is an opportunity to explain why, despite weakened westerlies, the rainfall at 6 ka over southern temperate Australia may have been equivalent to, or slightly higher than at 0 ka. Such information may be useful for providing alternative explanations for processes derived or inferred

from proxy data.

In the 6 ka simulations, insolation is higher at the southern pole in the annual mean and lower at low latitudes, relative to 0 ka (Fig. 2(b)). This change in insolation would, to first order, act to reduce the equator to pole temperature gradient. In order to identify whether the equator to pole temperature gradient has reduced, the following calculation is undertaken:

$$DT_{ep} = \bar{T}_{30Nto30S} - \bar{T}_{60Sto90S} \tag{1}$$

Where $T_{30Nto30S}$ is the area averaged surface temperature (K) between 30°N and 30°S, $T_{60Sto90S}$ is the area averaged surface temperature (K) between 60°S and 90°S and $DT_{ep}$ is the equator to pole temperature difference (K). The values of $DT_{ep}$ are calculated for each individual model and plotted in Fig. 12 for the 0 ka simulations (white box, left axis), the 6 ka simulation (amber box, left axis) and the difference in DTep for 6 ka relative to 0 ka (pink box, right axis). The median DTep





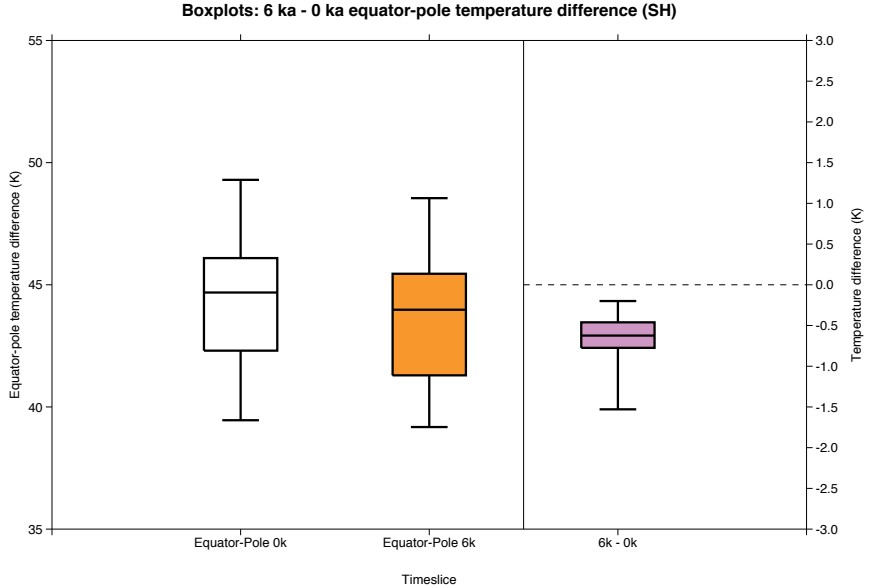

**Figure 12.** Box-and-whisker plots of the individual model mean difference in surface temperature between $30°N – 30°S$ and $60°S – 90°S$ in the 0 ka simulations (white boxes, left axis, K) and the 6 ka simulations (amber boxes, left axis, K). The pink box (associated with the right axis, K) is the individual model difference in the equator to pole temperature gradient for 6 ka relative to 0 ka.

from the models is 44.7 K for 0 ka and 44.0 K for 6 ka; however, all 32 models simulate a reduction in the difference in temperature between the equator and poles (pink boxplot?upper whisker is less than zero), which is an important result with respect to the strength of the westerlies.

With respect to the meridional temperature gradient, the thermal wind balance equation can be written as:

$$\frac{\Delta U_g}{\Delta z} = \frac{fT}{g}\frac{\Delta T}{\Delta y} \tag{2}$$

Where $\Delta U_g/\Delta z$ is the change in the zonal geostrophic wind ($\Delta U_g$ m s$^{-1}$) with height ($\Delta z$, m—also called the vertical shear of the geostrophic wind with respect to height), f is the coriolis parameter (s$^{-1}$), T is the temperature at a reference point (K), g is the acceleration due to gravity (m s$^{-2}$), and $\Delta T/\Delta y$ is the change in surface temperature (in this case, K) per distance of latitude (or the surface temperature gradient, m). Following Equ. 2, reducing equatorial surface temperatures and increasing them at high-latitudes would reduce the $\Delta T/\Delta y$ term. If all other parameters are held fixed then $\Delta U_g/\Delta z$ will also reduce.

10 Therefore, given the insolation and surface temperature characteristics of the PMIP simulations (lower insolation and surface temperature at low latitudes and vice-versa for high-latitudes), the westerly winds must therefore weaken in the mid-latitudes at 6 ka relative to 0 ka.





So, given that there is a physically plausible reason why the westerlies would have been weaker at 6 ka relative to 0 ka, why is there little change to the annual mean rainfall (or even a tendency for a small increase)? To answer this question, the seasonal cycles of insolation, temperature and precipitation are plotted for the 0 ka and 6 ka simulations in Figs. 13(a) – (c) and for 6 ka minus 0 ka in Fig. 13(d). At 0 ka and 6 ka the insolation peaks in December and is lowest in June; however, at 6 ka insolation is higher in July to November and lower in December to May than at 0 ka. There is also a shift in the seasonal surface temperature and local SST with higher surface temperatures at 6 ka relative to 0 ka between August to January. Between August and November there is lower convective precipitation and, between December to June, higher convective precipitation in the 6 ka simulations relative to 0 ka. This suggests that the non-convective rainfall (e.g. frontal rain) is not changing. Furthermore, in April to June, the convective precipitation has increased more than the total precipitation change, which indicates that the non-convective rainfall has actually reduced during those months. So, there has been an increase in convective rainfall in the PMIP models (either through increased frequency and/or intensity), which has compensated for any reduction in precipitation caused by the weaker westerlies. Therefore, the relationship between westerly wind strength and precipitation could break down through changes in convective precipitation (which can be directly diagnosed from the models) and highlights where model data may be useful for providing alternative explanations for past climatic states.

There is another important caveat associated with the model-derived precipitation estimates for 6 ka. The coarse resolution of the models means that surface topographical features on the land are not represented well. Therefore, the impact of such topography on the prevailing circulation and precipitation would also be misrepresented. Areas where such a problem may be important are over the Great Dividing Range and Tasmania. It is logical to conclude that the misrepresentation of topography may also be contributing to any proxy-model disagreement. The only way to resolve such an issue would be to run high-resolution regional climate model simulations over the Southern Temperate zone from fully coupled, multi-millennial transient global model simulations. A measure of the time-dependent change in the circulation and its interaction with the land surface could then be assessed. Such model simulations have been shown to improve the representation of present day precipitation over Southern Alps of New Zealand (Ackerley et al., 2012) and have also been applied to simulations of 6 ka (Ackerley et al., 2013).

## 4.3 Future directions

Given the assessment above, this section focuses on some key opportunities for future work from both the proxy and modeling communities in order to provide a better platform to undertake fully integrated studies.

### 4.3.1 Proxies

Due to the sampling resolution of most of the proxy records and the response time of the systems from which they come, it is very difficult to reconcile seasonal variability. Exceptions to this are coral and speleothem records, although their coverage of the vast continent of Australia is sparse. One area for improvement in proxy reconstructions is a clear understanding of the season that is represented by (particularly) the biological archives, e.g. the season of pollen production and dispersal or invertebrate blooms. In many cases this may be known for the organisms in question, but often not adequately described in





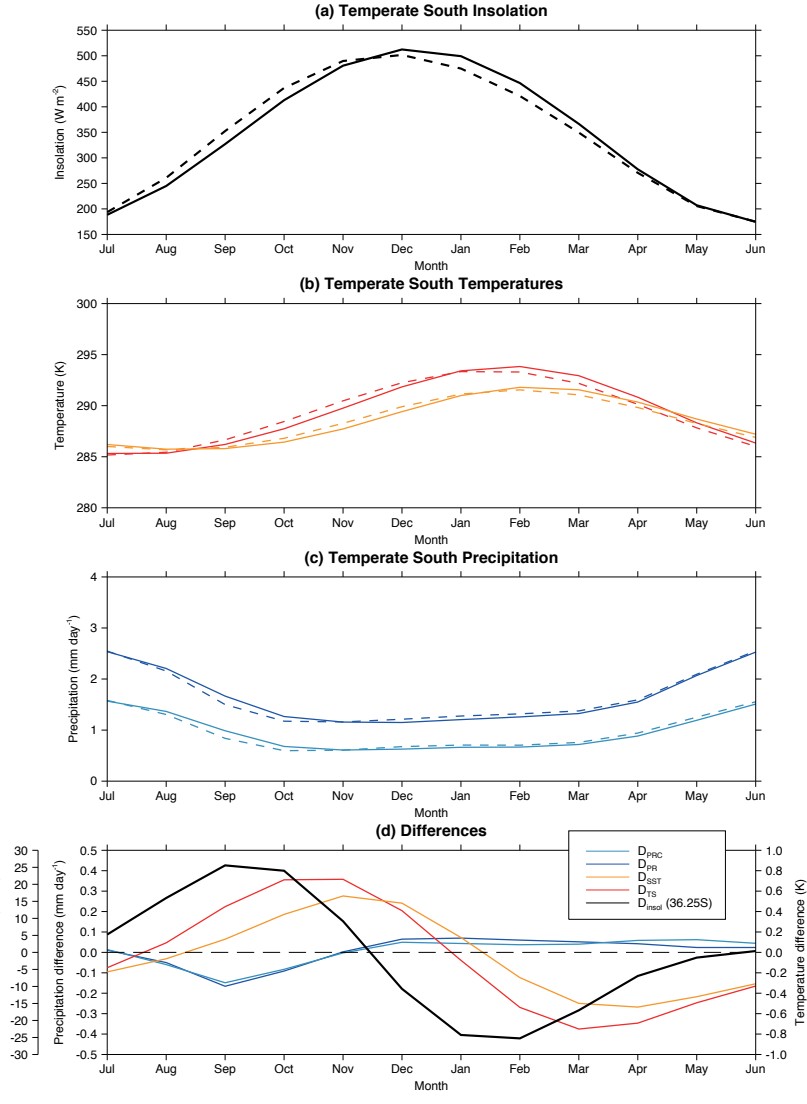

**Figure 13.** The monthly, ensemble and regional mean (a) insolation (taken at $36.25°$S for insolation, black line, W m$^{-2}$), (b) surface temperature (land and ocean combined, red line, K) and sea surface temperature (when available, amber line, K) and, (c) total precipitation (blue line, mm day$^{-1}$) and convective precipitation (turquoise line, mm day$^{-1}$) at 0 ka (solid lines) and 6 ka (dashed lines) within the Temperate South box. The difference in those fields (insolation, temperature and precipitation) for 6 ka minus 0 ka is plotted in (d).





the reconstructions, or the ranges not considered. There is great potential to re-interrogate the proxy records in view of the model outputs with regards changes in the seasonality of the signal. There are also possibilities of providing more quantitative data for proxies, either through the use of transfer functions or calibrating geochemical variability on the organisms directly, to look more at seasonal variability. Suitable proxies for these studies include tree rings, speleothems and molluscs that show

clear incremental growth in addition to coral records. As always, more robust chronologies can only benefit high-resolution palaeoclimatic work.

### 4.3.2   Models

The original work by Reeves et al. (2013a) compares the regional surface temperature and effective precipitation characteristics of one period relative to a previous one and not relative to the present day. Therefore, the first logical step would be to run time

slice simulations of each of the time periods discussed in Reeves et al. (2013a). A more ambitious plan would be to develop transient model simulations of the last 35 ka, which would allow a direct comparison with the OZ-INTIMATE synthesis; however, such simulations would be computationally very expensive. Despite that, there are some multi-millennial model simulations that have already been undertaken (e.g. Wagner et al., 2007; Menviel et al., 2011) and have also been planned as part of PMIP4 (Ivanovic et al., 2016), which may provide a template for other modelling groups to follow. Such simulations

would also be useful to investigate decadal to millennial variation in large-scale climate modes (such as ENSO). Lastly, in order to acquire high-resolution model data to compare with the proxies, dynamical downscaling of GCM simulations using Regional Climate Models could be employed (e.g. Ackerley et al., 2013). This may be particularly important over complex terrain (such as Tasmania and the Great Dividing Range) where this study has identified differences between the proxy reconstructions and modelled climates.

## 5   Summary and conclusions

This study aimed to apply the method outlined in Reeves et al. (2013a) to understand spatially coherent climate change signals for the mid-Holocene (6 ka) in an ensemble of climate models. Where the models and proxies agreed, the influence of the external forcing (insolation) or circulation (atmospheric dynamics) was presented in order to evaluate the proxy interpretation. Where there was conflict, the cause of the disagreement was presented and provides a point from which further investigation

can be undertaken.

Of the nine sub-regions (Fig. 1), all but two showed agreement with the proxy reconstructions. In the regions where the proxies and models agree, this study has shown that:

• In most of these areas, surface temperature and precipitation respond directly to the changes in insolation;

• The one exception was the tropical north-west where precipitation was driven by circulation change and not directly from the insolation; and,

• Southern Ocean SSTs are higher at 6 ka relative to 0 ka from the increased insolation (albeit weaker in models than suggested




by the proxies).

The two sub-regions that did not agree with the proxy reconstructions well were the tropical north-east and the temperate south. The reasons for the disagreement are:

• The simulated change in climate at 6 ka is sensitive to the "cold tongue bias" in the tropical Pacific apparent in the 0 ka simulations. It is the enhanced easterly flow over the tropical Pacific (from the stronger south-east Asian monsoon) that enhances the error;

• Annual mean rainfall over the temperate south is unchanged for 6 ka relative to 0 ka despite weaker westerly flow. Higher convective precipitation balances a reduction in precipitation from extratropical systems. Modern day analogues of relating

precipitation to westerly wind strength (i.e. positive correlation between wind strength and precipitation) may not be appropriate. Conversely, the coarse resolution of the GCMs may mean they are not representing the climate in topographically diverse regions (e.g. Tasmania and the Great Dividing Range).

Overall, this study shows that "upscaling" proxy reconstructions (to provide coherent regional information) is the most di-

rect method to compare with coarse-resolution climate models. Where there is agreement, the models can be used to verify the inferred dynamical processes from the proxies. Where there is disagreement, the proxies and models can be evaluated further to understand (and hopefully address) the cause of the mismatch. There is also scope for further integration of the Reeves et al. (2013a) method with model simulations by undertaking transient simulations of the Holocene as a whole (e.g. Wagner et al., 2007; Ivanovic et al., 2016) or other periods. Furthermore, there is also a clear need for acquiring seasonally resolved proxies

to evaluate the impacts of orbital variations on seasonality.

*Acknowledgements.* This project was funded by the ARC Centre of Excellence for Climate System Science (CE110001028). Steven Phipps was supported under the Australian Research Council's Special Research Initiative for the Antarctic Gateway Partnership (Project ID SR140300001). Cameron Barr was supported by the ARC Discovery Grant DP150103875. We acknowledge the World Climate Research Programme's Working Group on Coupled Modelling, which is responsible for CMIP, and we thank the climate modeling groups (listed in

Table XX of this paper) for producing and making available their model output. For CMIP the U.S. Department of Energy's Program for Climate Model Diagnosis and Intercomparison provides coordinating support and led development of software infrastructure in partnership with the Global Organization for Earth System Science Portals. The PMIP3 data were made available through the National Computational Infrastructure (NCI), which is supported by the Australian Government. We acknowledge the international PMIP2 modeling groups for providing their data for analysis, the Laboratoire des Sciences du Climat et de l'Environnement (LSCE) for collecting and archiving the model

data. The PMIP2/MOTIF Data Archive is supported by CEA, CNRS, the EU project MOTIF (EVK2-CT-2002-00153) and the Programme National d?Etude de la Dynamique du Climat (PNEDC).



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



**Table 2.** A list of the models used in this study, their original resolution / grid spacing and main references. The table is split into PMIP2 (1. – 18.) and PMIP3 (19. – 32.) models. For the PMIP2 models (1. – 18.), an asterisk (∗) denotes that the simulation is run with a dynamic vegetation scheme.

| Model acronym | Atmosphere resolution Horiz. grid [levels] | Ocean resolution Horiz. grid [levels] | Main references |
|---|---|---|---|
| **PMIP2** | | | |
| 1. CCSM3 | T42 [26] | 1° x 1° [40] | Otto-Bliesner et al. (2006) |
| 2. CSIRO-Mk3L-1.0 | R21 [18] | 5.625° x ∼3.18° [21] | Phipps et al. (2011, 2012) |
| 3. CSIRO-Mk3L-1.1 | R21 [18] | 5.625° x ∼3.18° [21] | Phipps et al. (2011, 2012) |
| 4. ECBILT-CLIO-VECODE | T21 [3] | 3° x 3° [20] | Renssen et al. (2005) |
| 5. ECBILT-CLIO-VECODE∗ | T21 [3] | 3° x 3° [20] | Renssen et al. (2005) |
| 6. ECHAM5-MPIOM1 | T31 [19] | 1.875° x 0.84° [40] | Roeckner et al. (2003); Marsland et al. (2003) |
| | | | Haak et al. (2003) |
| 7. FGOALS-g1.0 | 2.8° x 2.8° [26] | 1° x 1° [33] | Yongqiang et al. (2002, 2004) |
| 8. FOAM | R15 [18] | 2.8° x 1.4° [16] | Jacob et al. (2001) |
| 9. FOAM∗ | R15 [18] | 2.8° x 1.4° [16] | Jacob et al. (2001) |
| 10. GISSmodelE | 5° x 4° [12] | 5° x 4° [18] | Schmidt et al. (2006) |
| 11. UBRIS-HadCM3M2 | 3.75° x 2.5° [19] | 1.25° x 1.25° [20] | Gordon et al. (2000) |
| 12. UBRIS-HadCM3M2∗ | 3.75° x 2.5° [19] | 1.25° x 1.25° [20] | Gordon et al. (2000) |
| 13. IPSL-CM4-V1-MR | 3.75° x 2.5° [19] | 2.0° x 0.5° [31] | Marti et al. (2005) |
| 14. MIROC3.2 | T42 [20] | 1.4° x 0.5° [43] | K-1-Model-Developers (2004) |
| 15. MRI-CGCM2.3fa | T42 [30] | 2.5° x 0.5° [23] | Yukimoto et al. (2006) |
| 16. MRI-CGCM2.3fa∗ | T42 [30] | 2.5° x 0.5° [23] | Yukimoto et al. (2006) |
| 17. MRI-CGCM2.3nfa | T42 [30] | 2.5° x 0.5° [23] | Yukimoto et al. (2006) |
| 18. MRI-CGCM2.3nfa∗ | T42 [30] | 2.5° x 0.5° [23] | Yukimoto et al. (2006) |
| **PMIP3** | | | |
| 19. BCC-CSM1-1 | T42 [26] | 1° x 1/3°–1° [40] | Wu et al. (2010) |
| 20. CCSM4 | 1.25° x 0.9° [27] | 1.125° x 0.27-0.64° [60] | Gent et al. (2011) |
| 21. CNRM-CM5 | 1.875° x 1.875° [31] | 0.7° x 0.7° [42] | Voldoire et al. (2013) |
| 22. CSIRO-Mk3.6.0 | 1.875° x 1.875° [18] | 0.9° x 1.875° [31] | Rotstayn et al. (2012) |
| 23. CSIRO-Mk3L-1.2 | R21 [18] | 5.625° x 3.18° [21] | Phipps et al. (2011, 2012) |
| 24. EC-EARTH2.2 | T159 [62] | 1° x 1° [31] | Hazeleger et al. (2012) |
| 25. FGOALS-g2 | 2.8125° x 2.8125° [26] | 1° x 0.5°–1° [30] | Li et al. (2013) |
| 26. FGOASL-s2 | R42 [26] | 1° x 0.5°–1° [30] | Bao et al. (2013) |
| 27. HadGEM2-CC | 1.875° x 1.25° [60] | 1.875° x 1.25° [40] | Martin et al. (2011) |
| 28. HadGEM2-ES | 1.875° x 1.25° [38] | 1° x 1/3°–1° [30] | Martin et al. (2011) |
| 29. IPSL-CM5A-LR | 3.75° x 1.875° [39] | 2° x 0.5°–2° [31] | Dufresne et al. (2013) |
| 30. MIROC-ESM | T42 [80] | 1.4° x 0.5°–1.4° [44] | Watanabe et al. (2010) |
| 31. MPI-ESM-P | T63 [47] | 1.5° x 1.5° [40] | Giorgetta et al. (2013) |
| 32. MRI-CGCM3 | T159 [48] | 1° x 0.5° [50] | Yukimoto et al. (2012) |