# Peer review of "Evaluation of PMIP2 and PMIP3 simulations of mid-Holocene climate in the Indo-Pacific, Australasian and Southern Ocean regions"

_Climate of the Past, 2016_

## Referee Comment (RC1) · Anonymous Referee #1 · 18 Jan 2017

General:

The article presents a (comprehensive) summary of proxy and model based results for investigating changes between the mid-Holocene and pre-industrial period for western Pacific and adjacent regions. The authors use a large number of proxy data to compare with temperature and precipitation, derived from the Paleoclimate Model Intercomparison Program (PMIP2 and PMIP3). In the second part of the manuscript they try to explain model-data (mis-) matches by dynamical processes.

The concept and outline of the article look promising but a closer inspection of the manuscript reveals that the authors present a collection of (raw) GCM model output data and compare them one-to-one with results based on published proxy literature.

In addition, they i) apply and interpret results in an inconsistent way and ii) for most cases don't use the variety of large-scale output from the climate models to test their hypothesized physical explanations. Moreover, the different sections are not very well organized into distinct results and discussion sections making it difficult to follow.

Although this kind of data-model comparison is often used, new methodological approaches are available in the context of model-data comparisons. These include Proxy system models (Dee et al. 2016 (incl. source code), application of numerical/statistical downscaling to improve regional precipitation characteristics (Fallah et al., 2015; Wagner et al., 2012), applying climate field reconstructions using a coherent network of proxy data (PAGES2k, 2015) and using pseudo proxy experiments in the virtual world of climate models to test the spatial representativeness of single proxies or their networks (Smerdon et al., 2012). These methods should be used and applied or at least been mentioned in the context of state-of-the-art proxy-model comparisons.

In all, I cannot suggest publication of the manuscript in its present form. Below I tried to include a number of suggestions how to improve the manuscript for a revised version to address my main concerns and to put the conclusions of manuscript in context of different sources of errors involved in direct proxy-(multi-)model comparisons.

Specific:

Abstract:

For my taste the first paragraph of the abstract should be included into the introduction because it is not related to any scientific advancement achieved with the manuscript.

Specific:

Introduction:

p.2 l.6: Proxies are not observationally based. At least the wording should be changed e.g. "Proxies give indications of past climate changes albeit with uncertainties associated to their individual recorder characteristics on changes on meteorological variables

related to temperature/precipitation." same sentence: what's the rationale in comparing future simulations with reconstructions based on proxy data? I suggest re-phrasing the sentence that the comparisons may help to assess the general ability of models to simulate past climates which gives a certain degree of confidence in their ability to simulate potential future climate changes under specific forcing scenarios.

p.2 ll15 ff.: I suggest adding a few lines contrasting the merits/shortcomings of statistical versus numerical downscaling approaches. For instance the statistical downscaling assumes a constant large-scale/local scale relationship throughout time, whereas the numerical downscaling might account for those non-stationarities. Also numerical downscaling might be afflicted by the shortcoming that the driving GCM does not realistically simulate the large scale circulation and thus the RCM inherits errors from the GCM which in most cases cannot be compensated for.

p2. l. 23: Please add a short note what you understand by "effective precipitation" ?

p.2 ll. 28ff: How were those regions defined ? Are they purely related to some ad-hoc regionalization or have some statistical tools been applied to discriminate those temperate/hydrological different regions (e.g. Cluster analysis or some EOF-based approach). According to the figure 1 I assume however that the regions are rather related to some geographical lat-lon based criterion.

p. 3 l.3: In my opinion the authors should already indicate here that the simulated raw GCM precipitation is afflicted with a very high degree of uncertainty, especially over convective-active region and that current studies or compilations (e.g. IPCC) show large discrepancies between GCM-derived precipitation and observational/re-analysis/satellite derived precipitation. This would further motivate approaches to downscale GCM results with statistical/numerical approaches or/and use additional approaches (forward modelling) for proxy-model comparisons.

p.4 l. 1ff: I like the approach of developing seasonally resolved proxies. One might add here already some issues involved in addressing this point with the variety of proxies

that are used over the study region (e.g. corals vs. speleothems vs. tree rings) and there pros/cons for seasonally resolved reconstructions.

p4. l 6ff: Maybe the authors can also state here that a congruence/disagreement between proxy and models does not necessary mean that in the real world this must be the case. Given the high degree of proxy and model uncertainty both "outcomes" can be right but for the wrong reason. This is just a hint towards a more general view of model-proxy comparisons and the ultimate need for i) a sound basis for comparisons and ii) the consideration of sound and robust dynamical mechanisms controlling the models' and proxies' mean state (and variability) for different time periods (which is already mentioned by the authors).

General comment on the Introduction:

Admittedly, given the manuscript is already very long I still think given the vast literature on the comparison between mid-Holocene and pre-industrial in various studies at least those publications that are most relevant in this context should be summarized in one paragraph and linked to the present study. This is also important to give the reader who is not so familiar with the topic a general idea about the basic climatic changes (e.g. intensified monsoon) and potential driving mechanisms (e.g. changes in earth' obliquity). In its present shape the intro is largely focused on an international consortium (OZ-Intimate) and a method used by Reeves et al. that might not provide the full spectrum on research that has been achieved in the last decade or so.

2 Data, Analysis and external forcings

2.1 Proxy data

p.5 How did the authors treat the different sources of proxies concerning their temporal resolution, their individual dating uncertainties, their seasonal biases and their different recorder characteristics in terms of meteorological sensitivity ?

2.2 Model simulations and boundary conditions

p.5 ll7 ff.: How many years are used for each time slice ? is it the same for all simulations ? The authors should also add a paragraph related to the large bandwidth of models they use for comparisons and that a separation of models showing a good/bad performance in the simulation of present day climate or specific variables is not possible in the subsequent multi model-mean analyses anymore.

p.5 ll 15-25: The authors might try to shorten this discussion to the most relevant insolation changes.

p. 6 For completeness the authors should add a sentence that the calendar was not changed between the MH and PI due to the precession changes. For periods with large gradients in insolation that might influence to a certain degree results based on the Gregorian calendar and the one based on a discrimination of seasons related to solely astronomical considerations (e.g. the periods between solstice and equinox).

p. 6 ll 9ff: Please add a short information on the target grid the models were interpolated.

2.3 Post 1750 C.E. datasets

p.6 l. 10: The authors might think of changing the wording of the header to "Re-analysis data sets". In this context I also suggest that authors add a few words on the reliability of the according data sets, especially the HadISST data set dating back to 1870 concerning the data availability, quality and coverage. Again it might be important to stress the fact that those data sets are not observational data. For the generation of re-analysis data sets, meteorological data are used in an assimilation scheme integrated into a comprehensive numerical model.

2.4 Analysis p 7 ll 15ff: Concerning the multi-model mean, I basically cannot support the approach because it represents a physically unrealistic state. Instead, an individual treatment of the models should be envisaged for all stages of the analyses. However, I acknowledge that the authors try to amalgamate and synthesize this kind of spread

into "consensus" maps. A suggestion to further test the robustness of the model results is to use certain thresholds e.g. 2 times standard deviation for a more robust assessment not only in the sign of changes but also in terms of the magnitude. This is also motivated by statements further down that despite a certain agreement in the sign of the multi-model mean no statistical significant differences can be seen ( e.g. p.9 ll 4ff).

3 Model and proxy synthesis

3.1 Annual mean

3.1.1 Surface temperature

p. 7 l. 24: Do the authors refer to surface temperature or 2m-(Near) surface temperatures ?

p 7 l 30ff: In the context of statistical significance it is important to at least mention the difference between statistical significance at a certain level and the physical relevance. I assume that the authors use the multi-model mean to carry out the t-test between MH and PI that is based on the number of years they use for their analysis. Given the usually low variance in tropical latitudes in the annual cycle also very small absolute changes in T2m become statistically significant just due to construction of the test algorithm. I further assume that the authors also don t take into account the serial correlation (SC) of the data that might also influence the level of statistical significance as the presence of SC can (profoundly) change the number of degrees of freedom (cf. effective sample size).

p. 9 ll. 6ff: The authors compare here their quantitative model estimates with empirical reconstructions – how robust are results based on the proxy records mentioned or at least what are uncertainties implicitly included in the studies cited ?

3.1.2 Precipitation

p9 ll. 19 ff: Taking raw precipitation from GCM output should be avoided. To circumvent this issue statistical and/or numerical downscaling methods should be applied first.

**CPD**

Here I also think it's important to have a look at the individual performance of the different GCMs. The situation gets even more complicated when authors compare their (continental-scale) regions with local information from proxy data. In contrast to temperature changes, precipitation may occur at spatial scales that are much more heterogeneous compared to temperature.

p9, 32 ff: This comment is related to my last one for temperature – how robust are results given uncertainty in the proxy and their ability to really record (solely) precipitation changes? The following sections are a repetition of the first one with seasonal focus but with very weak or vague statements concerning the representativeness of the respective proxy towards the respective season.

I suggest that the authors re-structure the whole section 3 into one section and discuss results only for robust proxy-model comparisons. Also the comparison between circulation and proxy should be restricted to model based results and if possible for individual models in their relation to the individual temperature/precipitation (hydrological) changes.

4 Mechanisms responsible for agreement and disagreement

4.1 Model-proxy agreement

4.1.1 Tropical north-west (TNW)

p. 16 ll. 8ff. The authors state that "but the modeled change is not statistically significant—although 65% of the models simulate higher rainfall at 6 ka." – I am wondering why there is surprise given the large spread and the very weak coherence in the sign of the models. If anything, I would call this a very weak tendency towards higher rainfall. So I think it's really important to stress the effect and meaning of statistical versus physical significance in the context of the multi-model mean differences for the various parameters.

In all, I'm not really convinced in the usefulness looking for statistical significance given

the issues mentioned above related to absolute differences and the effect of the number of degrees of freedom on the level of significance.

p. 18 l. 6f: This comment relates to the one on page 16: In this context the authors state that "only 62%" of the models agree. In an earlier statement they state this is two thirds agreement and exploit this number being significant in terms of the consensus. There should at least be a consistent nomenclature when the authors speak about model agreement being "large" or "rather evenly distributed". There are more occurrences of the inconsistent use, particularly if the percentage lies between 60 and 70 %.

p. 18 l.9 ff: The authors try to explain the changes based on dynamical reasoning. Unfortunately, they don t use the simulated and model based output. Instead some vague mechanisms including effects of sea breeze is suggested which can by far not be simulated by the climate models the authors use.

4.1.3 The arid zone

I still don t understand why the authors don t use the models to test the physical consistence of the mechanisms they hypothesize based on proxy evidence. In addition, it's very hard to distinguish the actual results section from an alone standing discussion section or information that might be important to know earlier e.g. the shortcomings of PMIP2 and PMIP3 presented on p 24 ll 4ff and other model deficiencies.

4.2 Model-proxy conflict

p25, l. 31ff: I find it a quite strong argument that just because the authors see "more confidence" in the proxies the models are being flagged wrong. The authors use a very large bandwidth in the complexity of models ranging from EMIC-type to comprehensive Earth System Models. I would expect a more detailed discussion if one could discriminate differences in between the models concerning their ability to simulate ENSO and if so, whether there is conceptual/technical reasoning, for instance related to the resolution of the according ocean model.

p 26 ll1 ff: Here again the authors begin a discussion about model deficiencies that should be placed elsewhere (e.g. in the general introduction) but not in the results section. Moreover, a distinction on the different complexity levels and resolution of the models should be clearly taken into account in their evaluation.

Future directions:

This section mirrors the (mostly) conservative nature of the authors' team to maintain their strategies for future directions, neglecting innovative methods to robustly and consistently compare data and models. For instance, it does not make any difference running longer simulations with the same models on their performance – their biases will still remain, also for ENSO dynamics. Even dynamical downscaling can only improve results over regions where there is some confidence in the driving model to realistically simulate the large-scale circulation at the lateral boundaries. Also the mentioned calibration of proxies will often fail because of the coarse temporal resolution of the proxy and it's ability to record to a high degree meteorological entities. Even if it would be nice to have this at hand, most of proxies presented in the study are not suited given i) the short length and availability of meteorological observations over the study region for calibration and ii) the complexity (and partly inability) for the inverse modelling of meteorological data based on proxy data.

References:

Bijan Fallah, Ulrich Cubasch, Kerstin Prömmel and Sahar Sodoudi (2015), A numerical model study on the behaviour of Asian summer monsoon and AMOC due to orographic forcing of Tibetan Plateau, Climate Dynamics, DOI:10.1007/s00382-015-2914-5.

Dee, S. G., N. J. Steiger, J. Emile-Geay, and G. J. Hakim (2016), On the utility of proxy system models for estimating climate states over the common era, J. Adv. Model. Earth Syst., 8, doi:10.1002/2016MS000677.

PAGES 2k Consortium (2013): Continental-scale temperature variability during the

past two millennia (2013) Nature Geoscience, 6, 339-346, doi:10.1038/ngeo1797.

Smerdon, J.E. (2012), Climate models as a test bed for climate reconstruction methods: pseudoproxy experiments, WIREs Climate Change, 3:63-77, doi:10.1002/wcc.149.

Wagner, S., Fast, I., and F. Kaspar (2012): Climatic changes and forcing mechanisms between 20th century and pre-industrial times over South America in regional model simulations, Climate of the Past , 8, 1599-1620, doi:10.5194/cp-8-1599-2012

---

## Editor Comment (EC1) · A. Lorrey (Editor) · 28 Feb 2017

I openly declare ahead of this review that I know almost all of the authors of this paper personally. I've worked directly with many of them in the past on other published work and with some of them on previously funded projects. I have no personal interest conflicts with any of them.

An additional review was requested of me by CPD because we have been unfortunate to have only one solicited review for this paper. As such I submit this review here, but would like the authors to understand that final adjudication will be undertaken in consultation with the editors of CPD.

Overall, I thought this was a useful manuscript to put forward and it will be a welcome addition to the regional literature when completed. Attempts to reconcile Australian palaeoclimate data with simulations are not numerous. The region has a great number of complexities of climate drivers and processes that could be evaluated using model-proxy comparisons.

However, I have some concerns about how this paper has been presented though, or at least how a few things are couched, and with some of the visual content that is presented (or data not shown). I suggest it could be publishable in CP, but only if many minor and some major revisions were undertaken. Please see specific comments in the attached PDF.

First, the balance of the paper strengths lean heavily toward the analysis and descriptions of the climate model simulations. I'm elated that some basic physics of the climate system has been brought to bear with the incorporation of the discussion on the latitudinal gradient and geostrophic wind equations. There are some very interesting findings here for the model results, but I also cannot determine if any bias corrections were actually undertaken for the circulation patterns or if there is simply reference to identifying them using the post 1750CE data sets. Please make this more clear. Ahead of the following comments, I would also suggest the authors simply recast the use of the palaeo proxy data network (from Reeves et al., 2013a) as supporting field-based information that the models can be compared against (details why are stated below). Section 2.1 is also poorly written; it leads off with a description of INTIMATE and then Reeves et al. 2013 - and it seems very odd to me that for a SHAPE issue that there is no mention of that initiative anywhere, which has superseded INTIMATE in the Southern Hemisphere. I think it would be more contemporary to refine the aims of the model-proxy intercomparison in light of the stated goals of the SHAPE IFG - which are similarly stated on the SHAPE project website.

Second, there appear to be no real surprises to me in terms of the findings - the proxy data-model comparisons are elementary (mostly descriptive, but still very useful

and clearly-written). They are divided into sections that essentially show where the proxy-model comparisons work, and where they don't. I would greatly appreciate if the listing of proxy data derived from Reeves et al., 2013 (fundamental to supporting this work) was tabulated, including all metadata about location, type of archive, dating controls, seasonal sensitivity and signature for climate during the 6k interval are stated. The Reeves et al., 2013 paper is also mentioned as providing 'a method' but it does not do that in terms of integrating the data or providing a dynamical understanding of past variability or change with reference to a mean climate state. That particular work collected climate proxy records under certain criterion, and binned them into different geographic regions for Australia. If you adopt the spatial division of Reeves et al., 2013, and the data series used there in, it would be best to simply say so. There in, those geographic regions are somewhat arbitrarily ascribed; but I temper this comment by saying in reality there is good reason to have made those divisions. Just a bit more support and justification from modern climate studies that indicate there is a strong reason for the geographic divisions would go a long way to informing the readership. I believe that information can be easily obtained, and cited in the revised work. And better recognition that the real strength of Reeves et al., 2013 is the pre-selected proxy data that are 'regionally-representative'.

Third, the Reeves et al., 2013 depictions did not compare the past climate change signals to a common modern interval, but rather assessed the direction of change from one time step to the next. This limits a meaningful comparisons of the past patterns that are shown in Reeves et al., 2013 to the climate model simulations shown in this study. I realise there were probably data limitations in Reeves et al., 2013 that sent those authors down such a path, but it was identified as problematic early on (in discussions in Aus-INTIMATE). In this paper, it (and the pictures showing signals for different time slices) is advocated as 'presenting a new opportunity to integrate models with data'. At the risk of repeating myself, it does not: What it does is supply a series of pre-screened data and climate signals for the mid Holocene where assessing PMIP2 and 3 model signals may be undertaken. The authors have largely done this in a point-by-point

fashion; if the data from Reeves et al. has been further transformed, it is not clear how it was done. deeper understanding from a data integration would have been more meaningful; so I feel justified in mentioning this specific point here.

Fourth, I would also strongly encourage the authors to submit the data from Reeves et al., 2013 along with this paper, or provide a supplement with stable URLs where the data may be obtained. Sub-issues related to the points of viewing and assessing those data are: a. mapping of proxy signals onto the PMIP simulation outputs shown in Figures 3, 4, and 5. b. being able to observe the time series for each c. seeing how the 6k signatures compare to modern or pre industrial times.

Fifth, the scaling of the proxy signals so that they are compatible with the GCM signals is still unclear to me. this relates to point number 3. In using a tercile-based evaluation system of the proxy data, one needs to create a distribution for the data, with reference to a common interval (also the same interval used in the control run for the model simulations), then establish what the thresholds are for the terciles to obtain meaningful signals (warm, wet, cold, dry etc). that has not been clearly shown anywhere here ... and it cannot rely on antecedent work. Seeing the data and the new analysis are required for the descriptions of the proxies to be understood as factual.

Addressing the above comments, the more minor grammatical issues in the text, and recasting the paper toward the main strengths (modelling results and forcing mechanisms, supported by point data, rather than proxy-model intercomparison) would see this through. I'd also like to encourage the authors to evaluate their future work section and to try to be more broad with regard to proxy development, chronology evaluation and integrative approaches that could help future efforts bring models and proxies together - please see if that can also be done in a more refined manuscript.

I am happy if the authors would like to discuss any of this business directly with me. Best wishes, Drew

Please also note the supplement to this comment:
http://www.clim-past-discuss.net/cp-2016-136/cp-2016-136-EC1-supplement.pdf

[Figure]

**Supplement:**

[revised manuscript text omitted]

---

## Author Comment (AC1) · 14 Apr 2017

*General: The article presents a (comprehensive) summary of proxy and model based results for investigating changes between the mid-Holocene and pre-industrial period for western Pacific and adjacent regions. The authors use a large number of proxy data to compare with temperature and precipitation, derived from the Paleoclimate Model Intercomparison Program (PMIP2 and PMIP3). In the second part of the manuscript they try to explain model-data (mis-) matches by dynamical processes. The concept and outline of the article look promising but a closer inspection of the manuscript reveals that the authors present a collection of (raw) GCM model output data and compare them one-to-one with results based on published proxy literature.*

*In addition, they i) apply and interpret results in an inconsistent way and ii) for most cases don't use the variety of large-scale output from the climate models to test their hypothesized physical explanations. Moreover, the different sections are not very well organized into distinct results and discussion sections making it difficult to follow. Although this kind of data-model comparison is often used, new methodological approaches are available in the context of model-data comparisons. These include Proxy system models (Dee et al. 2016 (incl. source code), application of numerical/statistical downscaling to improve regional precipitation characteristics (Fallah et al., 2015; Wagner et al., 2012), applying climate field reconstructions using a coherent network of proxy data (PAGES2k, 2015) and using pseudo proxy experiments in the virtual world of climate models to test the spatial representativeness of single proxies or their networks (Smerdon et al., 2012). These methods should be used and applied or at least been mentioned in the context of state-of-the-art proxy-model comparisons. In all, I cannot suggest publication of the manuscript in its present form. Below I tried to include a number of suggestions how to improve the manuscript for a revised version to address my main concerns and to put the conclusions of manuscript in context of different sources of errors involved in direct proxy-(multi-)model comparisons.*

**Response:** First of all, the authors would like to thank the reviewer for their detailed analysis of our paper. The comments have been helpful in making us review what should and should not be included in the analysis (particularly around statistical significance, which is discussed below). The authors do feel however, that while the reviewer makes a valid point about the other options for model-data comparison, one of the real strengths of this work is the comparison with (as termed) "raw data". Furthermore, the techniques the reviewer points out are only applicable to "quantitative" data-model comparisons whereas the OZ-INTIMATE reconstructions (that we compare to here) are "qualitative", which is a key reason why we do not use the techniques the reviewer points to. Most importantly however, is that all of the errors we highlight—that are visible in comparison to the proxies—are also clear when the models are compared to the modern instrumental datasets. We are also careful to cite the papers that make such comparisons in the discussion section (instead of solely focusing on the past climate). That said, we can clearly see the importance of the literature the reviewer refers to and so we have considered each one individually and indicated where we should include it in a revised version of the manuscript. The authors have found the following observations:

1. Fallah et al. (2015) and Wagner et al. (2012): The Fallah et al. (2015) paper employs a method of downscaling whereby a fully coupled general circulation model (GCM) is run with T31 (~350 km grid spacing) resolution. The sea surface temperatures (SSTs) and sea ice concentrations (SICs) are then used from that first run to drive a higher resolution atmosphere-only GCM. The Wagner et al. (2012) paper forces a regional climate model (RCM, 0.44° grid spacing) over South America from a fully coupled GCM also. Both are examples of the dynamical downscaling we refer to in section 4.3.2 and so we would include these references in an updated section 4.3.2 for any revised manuscript. There are some issues however. For Fallah et al. (2015), the "high resolution model" uses T63 resolution (~150 km grid spacing) and this is not high enough to resolve the topography that they imply is important to their study. Furthermore, it does not appear that they bias correct the SSTs or SICs and so the model will be subject to the same circulation errors that

may arise from biases in the SST from the original driving model. This is also the same issue for the Wagner et al. (2012) paper, which uses data from the fully coupled GCM to force the RCM. We therefore propose to include the following in section 4.3.2 to account for this:

"Lastly, in order to acquire high-resolution model data to compare with the proxies, dynamical downscaling of GCM simulations using higher resolution global GCMs (e.g. Fallah et al., 2015) or Regional Climate Models could be employed (e.g. Ackerley et al., 2013; Wagner et al, 2012). This may be particularly important over complex terrain (such as Tasmania and the Great Dividing Range) where this study has identified differences between the proxy reconstructions and modelled climates. Nonetheless, efforts to bias correct the boundary data supplied by the driving GCM need to be included (not done in Ackerley et al., 2013; Fallah et al., 2015; Wagner et al, 2012), otherwise the simulations may only reproduce existing systematic errors at higher resolution.

2. Dee et al. (2016): This paper described the method of assimilating proxy data into a GCM (or, in this case, an "intermediate complexity GCM", i.e. simplified GCM). This approach is fascinating and an important area of model development for better model-proxy integration. The authors would therefore refer to it in section 4.3.2 for future model developments in a revised manuscript. Nevertheless, Dee et al. (2016) make an important statement at the end of the paper that is relevant to our work here (second to last paragraph of their paper):

"As discussed, our results suggest that reconstruction still improves for some proxy systems using nonlinear PSMs within detection and attribution, but these results may be dependent on the pseudo-proxy experimental design. Factors such as GCM complexity and structural model errors, water isotope physics scheme, proxy network distribution… or structural design of each PSM all contribute to a modelling framework, which may not be representative of nature. Thus, ongoing and future work towards real-proxy reconstructions must validate both the PSMs and the GCMs against observations".

The final sentence is important as our study is actually "validating GCMs (directly) against observations" by taking the inferred (but real) state of the climate from the proxies and comparing it to the actual state of the models. Using raw model data is paramount to this as it allows the systematic errors in the models to be presented clearly—and those errors are not just important in a paleoclimate framework as they are also noted in many other studies (which we are careful to cite). Adjusting the model with data assimilation will not act to solve the errors in the models themselves and, as Dee et al. (2016) indicate, direct comparisons like our study are therefore still crucial. Nonetheless, as stated above, the authors agree that our paper (and the underlying principle of data assimilation with proxies) should cite the Dee et al. (2016) work in section 4.3.2.

3. PAGES2k (2013): The PAGES2k summary provides a global and continental-scale assessment of the climate over the last ~2000 years. The paper primarily looks at whether global climate change is visible in the continental-scale regions and to what extent, by using regionally coherent proxy reconstructions. This is essentially what the OZ-INTIMATE synthesis did (Reeves et al., 2013b, and references therein) over the region we describe in our paper. The regional demarcations, justified in Reeves et al. (2013b), are based on spatially coherent signals in the proxy network. Therefore, our work already applies the method inferred by the reviewer. Furthermore, given the subtleties of the differences in insolation forcing at 6 ka relative to 0 ka (that result in a reduction in the seasonal cycle of the Southern Hemisphere), we would not expect a regionally-coherent signal in either the proxies or the models over the whole domain presented in Figure 1. Nevertheless, the PAGES2k network undertake their analysis using a coherent set of regional proxy data and as we are endeavouring to do the same by re-visiting the data collected as part of OZ-INTIMATE, it is fully justified to include reference to PAGES2k (2013) and we would do so in an updated introduction.

4. Smerdon et al. (2012): The use of pseudo proxies is described in the Smerdon et al. (2012) paper, and is again an important point raised by the reviewer. The authors will therefore include this reference in section

4.3.2 as another useful method of evaluating model data and integrating it with proxies. As with the other studies given above, there are issues with this method too, which justifies our analysis. For example, errors in the simulated base state of a GCM i.e. the Equatorial Pacific cold tongue bias (which is ubiquitous in GCMs) may be important for the representation of pseudo coral. Do we bias correct the model output or do we adjust the algorithm for calculating the pseudo proxy? Is the change in climate even realistic given such an erroneous base state? Using a pseudoproxy in this instance would not improve our understanding of the underlying model error and would only serve as a more complex method of arriving at the same answer. Or, if bias corrected, could hide the error in the first place or make the algorithm give spurious results (i.e. the calibration is done on a grossly incorrect state).

The reviewer's comments here are ultimately very important and we see the need to be a bit more ambitious/wide-ranging in the section on future modeling. Nonetheless, the use of "raw model" data is still highly important in comparing models and proxies as it does not hide the underlying errors in both datasets. Furthermore, it provides a discussion forum from which to improve their integration (i.e. by using the methods in the literature the reviewer cites). Our study provides an Australasian-centric starting point from which such developments could be made. It also highlights where models may be useful to provide insight into proxy interpretation (e.g. the increase in convective rainfall over southern Australia counteracting the reduction in rainfall from extratropical cyclones as the westerlies weaken) despite their limitations. Again, this case shows where the "raw data" are clearly useful and can help both the modeling and proxy communities.

One of the real strengths of this work is that we do not need complex data analysis techniques to show that clear and well-known biases in the models are equally clear in simulations of past climate. Trying to adjust the model output to improve them only acts to ignore the real issues in the models (e.g. the Equatorial Pacific cold tongue bias, Southern Ocean climate and cloud biases, Equator-to-pole temperature gradient and its impact on mid-latitude circulation

and the seasonal cycle in the tropics). If we adjust our data then that will undermine the important points raised by this paper, which show that the prominent biases in the contemporary climate simulations are having a significant impact on past-climate simulations (and their evaluation). No amount of statistical or dynamical downscaling is going to remove the cold tongue bias or the errors in the clouds over the Southern Ocean for example—only through a co-ordinated effort in model development. Finally, while there has been a detailed analysis of the proxy data over Australasia (Reeves et al., 2013b and the other OZ-INTIMATE special issue papers cited), no such synthesis has been undertaken for the models. This paper therefore provides an important climate model synthesis, which has been lacking for the Australasian region despite the extensive proxy data gathering effort. The authors would also like to draw the reviewer's attention towards the summary paper by Braconnot et al. (2012), which includes summaries of model simulated and proxy-inferred climatic conditions over many regions of the globe for 6 ka. Australasia is a glaring omission from this work and our paper would provide a first step in rectifying that omission (the authors will also refer to the Braconnot et al. paper and the lack of an Australian perspective in an updated introduction). Furthermore, other studies have evaluated data from PMIP simulations regionally over, for example, South America (Prado et al., 2013) and Europe (Brewer et al., 2007; Mauri et al., 2014) and this study provides the first instance (to our knowledge) of this being done over Australasia. Therefore, this paper is a timely and necessary piece of work from which the community can move forward in (perhaps) a more integrated manner into PMIP4. Without this work (which can be somewhat regarded as a benchmarking exercise for the models), the lack of a model evaluation study over Australasia will remain.

**Specific Points**

*Abstract: For my taste the first paragraph of the abstract should be included into the introduction because it is not related to any scientific advancement achieved with the manuscript.*

**Response:** The authors agree with the point the reviewer raises and would remove the first paragraph of the abstract. We do not agree that it should be in the introduction as it would cause repetition and should simply be removed.

*Introduction: p.2 l.6: Proxies are not observationally based. At least the wording should be changed e.g. "Proxies give indications of past climate changes albeit with uncertainties associated to their individual recorder characteristics on changes on meteorological variables related to temperature/precipitation." same sentence: what's the rationale in comparing future simulations with reconstructions based on proxy data? I suggest re-phrasing the sentence that the comparisons may help to assess the general ability of models to simulate past climates which gives a certain degree of confidence in their ability to simulate potential future climate changes under specific forcing scenarios.*

**Response:** The authors agree with the reviewer and will update the manuscript as suggested. Regarding the "comparing future simulations with reconstructions," this was not the intended point of the sentence and we can see it is incorrect. We would write the following to replace the first two sentences of the introduction: "Proxies give indications of past climatic conditions (albeit with uncertainties associated to their individual recorder characteristics and relating them to meteorological variables) and can be used to assess the ability of general circulation models (GCMs) to represent past climate states. Moreover, if past climate states can be reproduced adequately with GCMs, then there can be a degree of confidence in their ability to simulate future climate change."

*p.2 ll15 ff.: I suggest adding a few lines contrasting the merits/shortcomings of statistical versus numerical downscaling approaches. For instance the statistical downscaling assumes a constant large-scale/local scale relationship throughout time, whereas the numerical downscaling might account for those non-stationarities. Also numerical downscaling might be afflicted by the shortcoming that the driving GCM does not realistically simulate the large scale circulation and thus the RCM inherits errors from the GCM which in most cases cannot be compensated for.*

**Response:** The authors agree that these considerations are useful (and important) for undertaking statistical and / or dynamical downscaling; however, as such downscaling methods are not employed in this paper, such a discussion is unnecessary and would only act to lengthen the paper. Our point is simply that the work by Lorrey et al. (2007; 2008) provided a route for easier, and (importantly) more direct comparisons to be made between proxy reconstructions and model data. The power of our study is that we do not need to use complex methods to evaluate the models and proxies to focus on the main discrepancies between both datasets i.e. no amount of statistical or dynamical downscaling is going to remove the cold tongue bias or the errors in the clouds over the Southern Ocean. These errors can just be diagnosed directly from the models and the proxy evidence simply helps to confirm their presence in the past. We fully understand the reviewer's point here and so we will add a sentence to say the above (relating to downscaling) in any revised version.

p2. l. 23: Please add a short note what you understand by "effective precipitation" ?
**Response:** We will add:
"…(proxy-derived effective precipitation: the combined effect of total precipitation, evaporation, air flow and vegetation cover)… "

*p.2 ll. 28ff: How were those regions defined? Are they purely related to some adhoc regionalization or have some statistical tools been applied to discriminate those temperate/hydrological different regions (e.g. Cluster analysis or some EOF-based*

*approach). According to the figure 1 I assume however that the regions are rather related to some geographical lat-lon based criterion.*

**Response:** The regions are defined meteorologically along the following definitions (which are explained in Reeves et al., 2013b and references therein):

Tropical Northern: The perennially wet tropics, which are split between Indonesia (western) and Papua New Guinea (eastern)

Tropical South: Monsoon dominated, tropical rainfall zone with the "quasi-monsoon" (easterly dominated—Suppiah et al., 1992) and the "pseudo-monsoon" (low-level westerlies around the heat-low over north-west Australia, Suppiah et al., 1992; Berry et al., 2011; Ackerley et al, 2014).

Arid interior: Desert, dominated by anticyclonic flow, and little precipitation with no clear seasonal bias.

Temperate east: Easterly dominated, receiving rainfall throughout the year.

Temperate south: Westerly dominated, primarily receiving rainfall in austral winter but high precipitation throughout the year.

Southern Ocean: The Antarctic Circumpolar Current and associated oceanic salinity and temperature fronts, which divide across approximately 50°S. Also, highly influenced by the strength and location of the mean westerly flow (and associated weather systems).

Therefore, there are many abundantly clear and important meteorological/oceanographical reasons to choose the regions as they are. They also represent the zones where the proxies show some regional coherence (again, as outlined in the cited literature from which they are taken). Ultimately, divisions must be placed somewhere and there will always be cases for moving them or keeping them unchanged. As we wanted to extend the analysis of Reeves et al. (2013b, and references therein) to the models, we decided to choose exactly the same regions to be consistent. Changing them in this instance would mean our study is no longer consistent with theirs and our conclusions would be invalid in the context of the previous work. Nevertheless, we agree with the reviewer that Figure 1 makes it appear that we have chosen a simple "lat-lon" split, which is not the case. To address the reviewer's important point, we would aim to annotate the figure to include some of the meteorological / climatological

justification for those reasons so that the reader does not necessarily have to use the Reeves et al. (2013b) paper.

*p. 3 l.3: In my opinion the authors should already indicate here that the simulated raw GCM precipitation is afflicted with a very high degree of uncertainty, especially over convective-active region and that current studies or compilations (e.g. IPCC) show large discrepancies between GCM-derived precipitation and observational/reanalysis/satellite derived precipitation. This would further motivate approaches to downscale GCM results with statistical/numerical approaches or/and use additional approaches (forward modelling) for proxy-model comparisons.*

**Response:** The authors agree that there is uncertainty associated with the representation of convective precipitation in the GCMs; however, the first step of our analysis was to identify model agreement and disagreement. In the tropics the models and proxies largely agreed, where the thermally direct responses to the insolation changes (i.e. convection increases/decreases with higher/lower insolation) coincide well. In the instance of the tropical north-west domain, where the large-scale circulation is more important for driving seasonal rainfall, the models captured the seasonal cycle well (as described). The only tropical domain with a poor representation of the climate at 6 ka was in the tropical north-east, which is clearly related to the cold tongue bias. Therefore, while we accept that the representation of convection is highly problematical in GCMs (in particular over Australia—see Ackerley et al. 2014; 2015), it does appear to respond to the insolation-driven forcing described in this paper and so does not need direct discussion in this instance. Nevertheless, we intend to move the section describing the tropical north-west into supplementary material (see the "Restructuring" section at the end of the responses).

*p.4 l. 1ff: I like the approach of developing seasonally resolved proxies. One might add here already some issues involved in addressing this point with the variety of proxies that are used over the study region (e.g. corals vs. speleothems vs. tree rings) and there pros/cons for seasonally resolved reconstructions.*

**Response:** We agree with the reviewer that a key future focus of selecting proxies is a) select seasonally resolved – where possible or at the very least b) have a clear understanding of the seasonal signature influencing proxies – whether this be biological (pollen, chironomids, diatoms, etc.) or geomorphological (e.g. fluvial, glacier). This has not necessarily been well articulated in some of the earlier literature. Although there is an increasing effort to collect corals, speleothems, tree rings (of select species) and molluscs, the spatial coverage of these proxies will in no way represent the whole of Australia. Therefore other methods need to be employed – and understood. This is something we emphasise in the future work section.

*p4. L 6ff: Maybe the authors can also state here that a congruence/disagreement between proxy and models does not necessary mean that in the real world this must be the case. Given the high degree of proxy and model uncertainty both "outcomes" can be right but for the wrong reason. This is just a hint towards a more general view of model-proxy comparisons and the ultimate need for i) a sound basis for comparisons and ii) the consideration of sound and robust dynamical mechanisms controlling the models' and proxies' mean state (and variability) for different time periods (which is already mentioned by the authors).*

**Response:** The reviewer makes an excellent point here. We agree that the uncertainties in both the models and the proxies could lead to both giving the same outcome for different reasons. Nonetheless, we have been as careful as possible to consider the physical processes that occur in the real world and how they are simulated in the models. Given that the proxy data are evaluated thoroughly in the cited literature, the boundary conditions that are forced upon the model are well known (i.e. orbital changes and greenhouse gases), and the known PMIP/CMIP biases are openly considered, our conclusions and interpretations of the processes are drawn upon the best knowledge available to us presently. We also feel that the only way to answer the reviewer's point here is to undertake the work outlined in the "Future work" section, which this paper

provides an important reference point for (as no-one else has done such a model evaluation—to our knowledge—for the region under investigation).

*General comment on the Introduction: Admittedly, given the manuscript is already very long I still think given the vast literature on the comparison between mid-Holocene and pre-industrial in various studies at least those publications that are most relevant in this context should be summarized in one paragraph and linked to the present study. This is also important to give the reader who is not so familiar with the topic a general idea about the basic climatic changes (e.g. intensified monsoon) and potential driving mechanisms (e.g. changes in earth' obliquity). In its present shape the intro is largely focused on an international consortium (OZ-Intimate) and a method used by Reeves et al. that might not provide the full spectrum on research that has been achieved in the last decade or so.*

**Response:** The authors understand the reviewer's point here and the introduction was written to provide a simple (and short) overview and rationale. A lot of the background is covered in Section 2 (describing the models used) with respect to the impact of the orbital parameters. Moreover, a lot of the literature is cited in the discussion section where the results are discussed in detail relative to the research that has been undertaken over the last decade or more. It would be unnecessary to repeat it in the introduction, as it would only act to lengthen the paper. Finally, the introduction should not be a "literature review" of the last decade of work that pre-empts the contents of the paper. Instead, we have used it to explain our reasoning for doing the study we have done and then show the results we have found and explain them in the context of the available literature in the discussion section.

*2 Data, Analysis and external forcings*
*2.1 Proxy data p.5 How did the authors treat the different sources of proxies concerning their temporal resolution, their individual dating uncertainties, their seasonal biases and their different recorder characteristics in terms of meteorological sensitivity?*

**Response:** The papers selected for the OZ-INTIMATE compilation represent the 'best' – or most scientifically robust papers, as nominated by many the Australian palaeo-community over a series of workshops held 2010-2012. The criteria for inclusion was described as thus in Reeves et al., 2013a:

"The criteria for inclusion of records into this synthesis would ideally be: continuity through the last 35 ka, sound chronology, centennial-scale or better resolution and unambiguous and quantifiable palaeoclimate estimates. Although this has been achieved in many of the marine and speleothem records, their spatial coverage does not represent the greater Australian region. We have therefore chosen here to also include: high-resolution short-term records (e.g. corals), discontinuous geomorphic records (e.g. fluvial, lake shore, dune, glacier) where the interpretation is robust (see Fitzsimmons et al., 2013 for further discussion, which includes details on the limitations of using the dryland geomorphic proxies), as well as qualitative records (e.g. pollen), although noting the context of the site and the limitations of each record (Fig. 2)."

The interpretation of each of the records was accepted as that previously published. The limitations on each of the 'types' of proxy records are outlined in each of the regional papers (Reeves et al., 2013b; Bostock et al., 2013; Petherick et al., 2013; Fitzsimmons et al., 2013).

For this paper, we looked only at the time slice of 6ka compared with pre-industrial conditions, with a window of ±500 years for those records that have only centennial resolution.

A suggestion by the guest editor has been to make available the database of records included in the OZ-INTIMATE compilations. We agree that this provides a good opportunity to do this, should a time extension be granted, and will include sampling and dating resolution, proxy type and key indicator of each proxy used.

*2.2 Model simulations and boundary conditions*

*p.5 ll7 ff.: How many years are used for each time slice ? is it the same for all simulations ? The authors should also add a paragraph related to the large bandwidth of models they use for comparisons and that a separation of models showing a good/bad performance in the simulation of present day climate or specific variables is not possible in the subsequent multi model-mean analyses anymore.*

**Response:** Data are taken for the full 100 years of the 6 ka and 0 ka simulations for the PMIP2 models and also the full 100 years from the PMIP3 6 ka models. For the 0 ka PMIP3 simulations, the length of the averaging period was typically between 100-1000 years, depending on the length of the pre-industrial control run undertaken by the modelling institute. Given that the multi-decadal to multi-centennial averaging period for all models, the internally generated variability is unlikely to be significant in simulations of 100-1000 years. In the text we will state that all 6 ka experiments use 100 years of data as do the PMIP2 0 ka experiments. Furthermore, we will state that all PMIP3/CMIP5 0 ka data are used (and can vary from 100-1000 years depending on institution) to keep the analysis as impartial as possible. Regarding the large number of models used, we agree with the reviewer and would include the following:

"Due to the large number of models considered in this study, a detailed analysis of individual model performance is not considered. The specific details of the individual model performances are discussed in the relevant references given in Table 2".

*p.5 ll 15-25: The authors might try to shorten this discussion to the most relevant insolation changes.*

**Response:** We would change this paragraph to be (90 fewer words):
"In order to show the impact of the orbital parameter differences, the zonal-mean change in incoming solar radiation at the top of the atmosphere (insolation) is plotted in Fig. 2(a) for 6 ka relative to 0 ka. The 6 ka insolation is lower over much of the Southern Hemisphere (SH) between December to June and higher between August and November. The zonal-mean difference in

insolation over the whole year is plotted in Fig. 2(b). There is lower insolation between 10°N – 40°S and higher insolation southward of 50°S. In Fig. 2(c) and (d), respectively, the insolation is split into two six-month seasonal means, which coincide with the times of year when the highest insolation (October to March) and lowest insolation (April to September) occurs in the SH. Between October to March (Fig. 2(c)), the zonal mean insolation is lower at 6 ka between 10°N – 50°S and higher southward of 65°S. Conversely between April and September the insolation is higher at all latitudes between 10°N – 90°S (6 ka relative 0 ka)."

*p. 6 For completeness the authors should add a sentence that the calendar was not changed between the MH and PI due to the precession changes. For periods with large gradients in insolation that might influence to a certain degree results based on the Gregorian calendar and the one based on a discrimination of seasons related to solely astronomical considerations (e.g. the periods between solstice and equinox).*

**Response:** The authors will include the following in the model simulations section (2.2):

"The calendar and seasonal definitions are the same in both the 0 ka and 6 ka simulations, as described in Braconnot et al. (2007)."

*p. 6 ll 9ff: Please add a short information on the target grid the models were interpolated.*

**Response:** The sentence will in an updated manuscript will read, "…common longitude-latitude grid (2.5°x2.5°) before undertaking the analysis…".

*2.3 Post 1750 C.E. datasets p.6 l. 10: The authors might think of changing the wording of the header to "Re-analysis data sets". In this context I also suggest that authors add a few words on the reliability of the according data sets, especially the HadISST data set dating back to 1870 concerning the data availability, quality and coverage. Again it might be important to stress the fact that those data sets are not observational data. For the generation of reanalysis data sets, meteorological data*

*are used in an assimilation scheme integrated into a comprehensive numerical model.*

**Response:** We are fully aware that reanalysis data are not observations and do not infer that in the paper at all. Furthermore, we think the current title is more clear for telling the reader that we are using post-1750, which is important for comparing with the 0 ka simulations (i.e. 1750 CE climate). Regardless of the possible biases in the reanalyses, they still represent the only observationally constrained estimate of the state of the atmospheric available (accepting we could choose one of several datasets). A discussion of the reliability is unnecessary as their internal uncertainty is unlikely to be important relative to the large errors in the models that we show (i.e. the tropical Pacific easterly bias leading to the cold tongue bias). The cold tongue bias is manifested by both the SST (negative anomalies) and circulation (easterly anomalies) errors in the models, which are confirmed by comparing the PMIP models with the two independent reference datasets (i.e. HadISST and ERA-Interim). It is highly unlikely that both ERA-Interim and HadISST datasets will be biased in a manner that gives the same result. Our analysis clearly shows that, whatever the biases are in the HadISST and ERA-Interim datasets, the strong and significant differences between the models and those datasets is a clear sign of a problem with the models and not the reference datasets. To that end we will include the following sentence in the discussion (to be added in the current discussion paper line 20, page 26 to follow on from the end of that paragraph):

"While there are uncertainties associated with both the HadISST2 and ERA-Interim datasets (see Rayner et al., 2003; Dee et al., 2011, respectively), the consistency between the SST (negative anomalies) and circulation (easterly anomalies) suggest that the biases in the models are more important than any uncertainty in the reference datasets."

*2.4 Analysis p 7 ll 15ff: Concerning the multi-model mean, I basically cannot support the approach because it represents a physically unrealistic state. Instead, an individual treatment of the models should be envisaged for all stages of the analyses. However, I acknowledge that the authors try to amalgamate and*

*synthesize this kind of spread into "consensus" maps. A suggestion to further test the robustness of the model results is to use certain thresholds e.g. 2 times standard deviation for a more robust assessment not only in the sign of changes but also in terms of the magnitude. This is also motivated by statements further down that despite a certain agreement in the sign of the multi-model mean no statistical significant differences can be seen ( e.g. p.9 ll 4ff).*

**Response:** While the authors understand the reviewer's reservations about the multi-model mean (MMM), this statistic still remains one of the best ways of showing the mid-point of a large ensemble of models. Furthermore, our method of showing the "consensus" is an attempt to account for whether the mean value is representative of most of the models. This is something that was used extensively in the IPCC AR5 (Flato et al., 2013; Collins et al., 2013; Chapters 9 [Figs. 9.2–9.5] and 12 [Figs. 12.11 onwards], respectively). As the reviewer suggests, another approach is to use a different threshold, such as 2 times the standard deviation; however, using such an approach is unlikely to change the outcome of our analysis (the 2 times standard deviation is very likely to give a spread that is similar to our given confidence intervals). We have chosen to provide a synthesis that gives the reader a measure of the mid-point, its statistical significance and the agreement of the sign of the change relative to the mean in the control state. The authors do agree with the reviewer fully on the language used around the statistical interpretation, however. We will re-structure the paper to focus on the changes that have both statistical and (more importantly) physical relevance. We refer the reviewer to our planned changes in the "Restructuring" section of our response.

*3 Model and proxy synthesis*

*3.1 Annual mean*

*3.1.1 Surface temperature p. 7 l. 24: Do the authors refer to surface temperature or 2m-(Near) surface temperatures ?*

**Response:** The authors are referring to surface temperature so that it provides a continuous transition from sea surface to land surface temperatures. Given that

the proxies do not infer air temperature at a specific height above the ground and that there are variable amounts of land and ocean coverage in the analysis regions, we chose to use the surface values in this instance (as stated already in the text).

*p 7 l 30ff: In the context of statistical significance it is important to at least mention the difference between statistical significance at a certain level and the physical relevance. I assume that the authors use the multi-model mean to carry out the t-test between MH and PI that is based on the number of years they use for their analysis. Given the usually low variance in tropical latitudes in the annual cycle also very small absolute changes in T2m become statistically significant just due to construction of the test algorithm. I further assume that the authors also don t take into account the serial correlation (SC) of the data that might also influence the level of statistical significance as the presence of SC can (profoundly) change the number of degrees of freedom (cf. effective sample size).*

**Response:** The authors understand the reviewer's point here and we fully agree that physical relevance is more important than the statistical significance. We try to use the statistical test (in combination with the consensus) to provide a basis from which to explore the physical implications (which we try to do in the discussion). As the reviewer correctly points out, interannual temperature variability in the tropics is generally low and therefore a small change can result in a "statistically significant result". Nonetheless, in the case of the tropical zones, it is clear that overall mean insolation was lower at 6 ka relative to 0 ka so there is already a very clear explanation for lower temperatures. Conversely, when the proxies indicate warmer conditions in the tropical north-east at 6 ka—but we only see cooler conditions—it cannot be an insolation-driven feature (if it were then the proxies would indicate cooler conditions too). This is why we present the cold tongue bias issue. We would expect slight cooling from the insolation changes but there should be a dynamically driven feature (e.g. the atmosphere-ocean circulation driven Indo-Pacific Warm Pool) that should be reversing the sign of the change. Given the combination of 81% (26/32) of the models simulate a cooling over the tropical north-east, the cold-tongue bias is a highly common

feature in the GCMs and, the associated mean circulation changes/biases shown in Figure 11, we have a clear set of physical processes that give a statistically significant result that highlight the models' common failing. The authors therefore feel that our use of statistical significance is purely to give a starting point from which to base a physical interpretation (and not the conclusion). The authors accept there are several places with ambiguous descriptions of the statistical results but our aim here was to solely present all the analysis to the reviewer/reader. We have provided a summary (see the section at the end entitled "Restructuring") to show the reviewer where we have taken out less significant analyses in order to focus on the places where we have both good model and proxy coverage.

*p. 9 ll. 6ff: The authors compare here their quantitative model estimates with empirical reconstructions – how robust are results based on the proxy records mentioned or at least what are uncertainties implicitly included in the studies cited?*

**Response:** The SST records are some of the best resolved of the proxy models and have both high resolution dating and sampling. The analysis here is based on quantitative reconstructions of Mg/Ca and $\delta^{18}O$ of foraminiferal calcite. Where different organisms (i.e. forams that grow in different sections of the water column) and/or geochemistry has been utilised, this has been taken into consideration in the reconstruction (Reeves et al., 2013b). The typical error for these types of reconstructions is ±1 K (Helen Bostock—personal communication).

*3.1.2 Precipitation p9 ll. 19 ff: Taking raw precipitation from GCM output should be avoided. To circumvent this issue statistical and/or numerical downscaling methods should be applied first. Here I also think it's important to have a look at the individual performance of the different GCMs. The situation gets even more complicated when authors compare their (continental-scale) regions with local information from proxy data. In contrast to temperature changes, precipitation*

*may occur at spatial scales that are much more heterogeneous compared to temperature.*

**Response:** Statistical or dynamical downscaling of the raw data would only result in us attaining the same conclusions (and analysis) except the data will simply be presented at a higher resolution (i.e. higher resolution will not improve the result). As the reviewer correctly points out later in their review (reference to the future work section):

"*Even dynamical downscaling can only improve results over regions where there is some confidence in the driving model to realistically simulate the large-scale circulation at the lateral boundaries,*"

This statement seems somewhat contradictory to the reviewer's point above by suggesting that we employ downscaling here. It is important to re-iterate that the raw data are useful in this study. The models simulate the production of precipitation using a known set of resolved or parametrized physical processes. Therefore, if we do not plot the simulated (raw) precipitation data we cannot infer the likely processes (dynamics) that govern the precipitation produced by the models (and the associated biases). Furthermore, in this instance, verifying the main features across the whole ensemble is important as it also highlights the well-known errors in model simulations in this region.

Regarding individual model performances: while it may be interesting to see which models produce the best climate on a region-by-region basis it is not a good way of evaluating model performance. What if one model represents the climate in one of our regions well but performs the worst in all other regions? What if that same model also has the worst simulation of the global climate? Do we investigate it further in that region based on it doing well locally or do we reject it? In our analysis, an evaluation of the overall model consensus is important as it helps us maintain a broader view of model errors across the ensemble. Evaluating individual models in fine detail will lengthen the paper considerably but would be a useful area of future work for another paper. Finally, the authors accept that the proxies are taken at point locations; however, the work undertaken by Reeves et al. (2013b—and references therein) was

careful to choose proxies that produced regionally coherent climatic signals across the regions we have investigated (and was made clear in those papers).

*p9, 32 ff: This comment is related to my last one for temperature – how robust are results given uncertainty in the proxy and their ability to really record (solely) precipitation changes? The following sections are a repetition of the first one with seasonal focus but with very weak or vague statements concerning the representativeness of the respective proxy towards the respective season. I suggest that the authors re-structure the whole section 3 into one section and discuss results only for robust proxy-model comparisons. Also the comparison between circulation and proxy should be restricted to model based results and if possible for individual models in their relation to the individual temperature/precipitation (hydrological) changes.*

**Response:** The authors fully understand and agree with this point and we have provided a method for restructuring the paper at the end of the responses to account for the reviewer's comment. We refer the reviewer to that for answering this point. If the Editorial staff allow us to submit a revised version of the manuscript then Section 3 will be changed in the way described in the "Restructuring" section at the end of these responses. It is important to note that we can only use qualitative language (e.g. wet/dry) at this stage with the available data (primarily from speleothems and their coverage is poor within the domain of interest). Such quantification, as the reviewer suggests, is an area of future research and efforts are being made (with isotopoic and hydrologic balance work) to address some of quantification issues but they are not available yet and so the assessment we provide is the best that can be done at present.

*4 Mechanisms responsible for agreement and disagreement*
*4.1 Model-proxy agreement*
*4.1.1 Tropical north-west (TNW) p. 16 ll. 8ff. The authors state that "but the modeled change is not statistically significant, although 65% of the models simulate higher rainfall at 6 ka." – I am wondering ˇ why there is surprise given the large spread and the very weak coherence in the sign of the models. If anything, I*

*would call this a very weak tendency towards higher rainfall. So I think it's really important to stress the effect and meaning of statistical versus physical significance in the context of the multi-model mean differences for the various parameters. In all, I'm not really convinced in the usefulness looking for statistical significance given the issues mentioned above related to absolute differences and the effect of the number of degrees of freedom on the level of significance.*

**Response:** Again, the authors fully see this point and agree with it. We feel that the TNW section provides a useful summary for readers but it is not necessary in the main text. We have suggested including it in a Supplementary Material section (see "Restructuring" statement at the end). This would remove the ambiguity around our statements about the statistics but allow readers to review the seasonal cycle changes between 6 ka and 0 ka in the models.

*p. 18 l. 6f: This comment relates to the one on page 16: In this context the authors state that "only 62%" of the models agree. In an earlier statement they state this is two thirds agreement and exploit this number being significant in terms of the consensus. There should at least be a consistent nomenclature when the authors speak about model agreement being "large" or "rather evenly distributed". There are more occurrences of the inconsistent use, particularly if the percentage lies between 60 and 70 %.*

**Response:** The authors fully agree with this and we will address this ambiguity when restructuring the paper (see the "Restructuring" section at the end of these responses). We will be careful to only draw attention to physically robust signals in the 6 ka simulations relative to the 0 ka.

*p. 18 l.9 ff: The authors try to explain the changes based on dynamical reasoning. Unfortunately, they don t use the simulated and model based output. Instead some vague mechanisms including effects of sea breeze is suggested which can by far not be simulated by the climate models the authors use.*

**Response:** The authors can clearly see the issue the reviewer is raising and why confusion may arise. We cite the Birch et al. (2015) paper as it highlights the importance of the sea breeze circulation for initiating precipitation in the real world; however, using that reference makes it seem like we are suggesting the models are representing something they clearly cannot. Nevertheless, the GCMs we analyse almost certainly use the hydrostatic approximation and therefore, to first order, an increase in the land temperatures over the continent will induce a reduction in the low-level density, ascending air over the land mass and subsequent convergent flow from the ocean to the land. This is why we state, "...the higher continental temperatures (September to December, 15 Fig. 7(b)) are likely to have enhanced onshore flow..." as it is consistent with the physics used in the models. Furthermore, as plotted in Figure 7, the land surface temperature increases, the land-based convective precipitation also increases and therefore (through continuity) there must be onshore flow to account for the vertical mass flux, which is caused by activating the convection scheme. Finally, stronger onshore flow can also be seen in Figures 3(c), 4(c) and 5(c), which is again consistent with the scientific reasoning presented here. We therefore feel that a simple re-wording of the paragraph is necessary here, which would be:

"The cause of the higher rainfall at 6 ka (particularly in October to March) over tropical south-east can be seen when the seasonal cycle of precipitation is considered (Fig. 7). Convectively generated precipitation and onshore flow, driven by the high (austral) summertime insolation, are important processes that govern precipitation over the Cape York Peninsula (i.e. north-east Australia, see Birch et al., 2015). The higher insolation in June to December (Figs. 7(a) and (d)) causes SST at 6 ka to be higher in August to January (Figs. 7(b) and (d), SST response lags the insolation change), which coincides with the period where both the convective and total precipitation are higher in the 6 ka simulations relative to 0 ka (Fig. 7(c) and (d)). Furthermore, higher land surface temperatures also coincide with the higher convective precipitation at 6 ka relative to 0 ka (Fig. 7(d)), which causes an increase in low-level convergence over the land that is consistent with the stronger easterlies (see Figs. 3(c), 4(c) and 5(c)). The earlier onset of the monsoon from increased continental heating, stronger onshore flow and higher SST adjacent to the land (i.e. higher

evaporation) is therefore likely to be responsible for the higher precipitation over the tropical south-east at 6 ka in the models. Conversely, the impact of reduced land and sea temperatures from April to July has little impact on precipitation during the dry season."

*4.1.3 The arid zone I still don t understand why the authors don t use the models to test the physical consistence of the mechanisms they hypothesize based on proxy evidence. In addition, it's very hard to distinguish the actual results section from an alone standing discussion section or information that might be important to know earlier e.g. the shortcomings of PMIP2 and PMIP3 presented on p 24 ll 4ff and other model deficiencies.*

**Response:** We feel that we do this adequately as we refer to the changes in insolation and the subsequent response of both surface temperature and convective precipitation to those changes. It is also clear that the increase in insolation from September to November is driving the increase in precipitation—as it is primarily convective precipitation that responds (i.e. the thermally direct response). Conversely, the reduction in December–March insolation corresponds with a reduction in convective precipitation (i.e. thermally direct response). The magnitude of the reduction of precipitation in January to March is larger than that of the September to November increase, which is consistent with the proxies suggesting a precipitation reduction in the northern half of the arid zone (as we state). Again, in the southern arid zone there is an increase in October to December precipitation that is primarily from an increase in convective rainfall (direct response to increased solar radiation). Given the weakening of the westerlies, it seems unlikely that the increase in precipitation is from mid-latitude systems; however, in order to diagnose this, a cyclone-tracking algorithm would need to be performed on all model data. That is beyond the scope of this synthesis paper and should be an area of future work and we will refer to that in the future work section. Finally, regarding the section wording—we will move much of the "results" parts of the discussion into the results (outlined in the "Restructuring" statement at the end of the responses) to shorten the paper and focus only on the seasonal cycle and convection in this

section. Nevertheless, the discussion of the model shortcomings should remain in the discussion. The discussion is there to discuss the results in the context of known processes, which includes an acknowledgement of those processes in the literature. An earlier discussion of the PMIP2/3 models and their limitations before this point would further (and unnecessarily) lengthen the paper.

*4.2 Model-proxy conflict p25, l. 31ff: I find it a quite strong argument that just because the authors see "more confidence" in the proxies the models are being flagged wrong. The authors use a very large bandwidth in the complexity of models ranging from EMIC-type to comprehensive Earth System Models. I would expect a more detailed discussion if one could discriminate differences in between the models concerning their ability to simulate ENSO and if so, whether there is conceptual/technical reasoning, for instance related to the resolution of the according ocean model.*

**Response:** This study only uses 1 EMIC-type model (ECBILT). All others are ESMs/GCMs and therefore the one EMIC should not have a significant impact on the results as it is considered in the context of all the other models. More importantly however, it is clear from the literature (as cited in the paper) that the models also share characteristic errors that occur in both CMIP3 and CMIP5 i.e. cloud errors over the Southern Ocean and the cold tongue bias in the Tropical Pacific. We simply use the previous literature to help provide an explanation for the disagreement between the models (known errors) and the proxies (which we have more confidence in in these instances). An investigation into the representation of ENSO in individual models is a separate piece of work that should standalone and is beyond the scope of our paper (and could be future work). Furthermore, the investigation of ENSO in the individual models is already provided through the papers we cite in the text. Ultimately, our conclusion is that the errors discussed widely in the literature are actually made worse in the mid-Holocene. The error enhancement of the cold tongue bias is caused by the stronger south-east Asian monsoon (driven by the insolation changes) acting to strengthen the existing anomalously strong equatorial Pacific easterlies in the models.

*p 26 ll1 ff: Here again the authors begin a discussion about model deficiencies that should be placed elsewhere (e.g. in the general introduction) but not in the results section. Moreover, a distinction on the different complexity levels and resolution of the models should be clearly taken into account in their evaluation.*

**Response:** The authors feel strongly that the place to discuss any literature relevant to the modelled processes we describe is in the discussion. If we were to incorporate the discussion as part of a general "literature review" in the introduction then it would lengthen the paper considerably. By the time a reader reaches the discussion section we would have to repeat the evaluation from the introduction again in to provide context for the results we present. Such repetition is unnecessary. We feel that an introduction should only introduce the background to the work undertaken (and its rationale) but should not pre-empt the physical processes that are highlighted in the discussion. By discussing it in the introduction we would be explaining the results before actually showing them. Instead, we present the results and then use the literature (and other figures) to explain them. Whether the reader encounters it in the introduction or the discussion is purely a matter of style.

*Future directions: This section mirrors the (mostly) conservative nature of the authors' team to maintain their strategies for future directions, neglecting innovative methods to robustly and consistently compare data and models. For instance, it does not make any difference running longer simulations with the same models on their performance – their biases will still remain, also for ENSO dynamics. Even dynamical downscaling can only improve results over regions where there is some confidence in the driving model to realistically simulate the large-scale circulation at the lateral boundaries. Also the mentioned calibration of proxies will often fail because of the coarse temporal resolution of the proxy and it's ability to record to a high degree meteorological entities. Even if it would be nice to have this at hand, most of proxies presented in the study are not suited given i) the short length and availability of meteorological observations over the*

*study region for calibration and ii) the complexity (and partly inability) for the inverse modelling of meteorological data based on proxy data.*

**Response:** The authors understand the points the reviewer raises and we would like to include the following reasoning for our suggestions. The reasons we chose our future directions to be are:

1. Running longer simulations will bring the models in line with other methods of evaluating past climatic states. For example, comparing climatic conditions between consecutive periods e.g. compare 21 ka with 14 ka, 9 ka with 14 ka, 5 ka with 9 ka etc. (as done in Reeves et al., 2013b). In running a model transiently through these time periods we can do an equivalent analysis and perhaps separate out the processes the models can represent well from those that they cannot.

2. For our suggestion to run high-resolution model simulations, we were primarily thinking about improving our ability to evaluate climatic change in mountainous regions that are sensitive to the direction of the incident flow and its strength (e.g. Tasmania and the Great Dividing Range). Such simulations would help us to see how the Southern Hemisphere westerlies (or other processes, e.g. convection) impact on past climate in topographically diverse regions. We could also bias correct the forcing model to improve the simulated climate in the RCM, which should negate some of the problems the reviewer describes.

3. Both this exercise and the Aus2k efforts have highlighted that although we have some excellent palaeo-records, they are held back by the poor dating resolution and/or lack of seasonally-resolvable signatures. In many cases resampling the sites or re-analysing the records could quickly address this.

Overall, we do see that maybe we were a little conservative in our suggestions and so we intend to extend the future work section considerably in a revised version of the manuscript to incorporate the suggestions from the reviewer. We will also incorporate the discussion of the new literature that the reviewer has suggested with respect to the modelling (by using some of the discussion presented at the start of our responses with respect to those papers).

**References**

Ackerley, D., Berry, G., Jakob, C. and Reeder, M. J. (2014), The roles of diurnal forcing and large-scale moisture transport for initiating rain over northwest Australia in a GCM. Q.J.R. Meteorol. Soc., 140: 2515–2526. doi:10.1002/qj.2316

Ackerley, D., Berry, G., Jakob, C., Reeder, M. J. and Schwendike, J. (2015), Summertime precipitation over northern Australia in AMIP simulations from CMIP5. Q.J.R. Meteorol. Soc., 141: 1753–1768. doi:10.1002/qj.2476

Berry, G., Reeder, M. J. and Jakob, C. (2011), Physical mechanisms regulating summertime rainfall over Northwestern Australia. J. Clim. 24, 3705–3717.

Bostock, H., Barrows, T., Carter, L., Chase, Z., Cortese, G., Dunbar, G., Ellwood, M., Hayward, B., Howard, W., Neil, H., Noble, T., Mackintosh, A., Moss, P., Moy, A., k, White, D., Williams, M., and Armand, L.: A review of the Australian–New Zealand sector of the Southern Ocean over the last 30 ka (Aus-INTIMATE project), Quaternary. Sci. Rev., 74, 35–57, 2013.

Braconnot, P., Harrison, S. P., Kageyama, M., Bartlein, P. J., Masson-Delmotte, V., Abe-Ouchi, S., Otto-Bliesner, B. and Zhao, Y., (2012), Evaluation of climate models using palaeoclimatic data, Nat. Clim. Chg., 2, 417–424.

Brewer, S., Guiot, J. and Torre, F. (2007), Mid-Holocene climate change in Europe: a data-model comparison, Clim. Past, 3, 499–512.

Collins, M., R. Knutti, J. Arblaster, J.-L. Dufresne, T. Fichefet, P. Friedlingstein, X. Gao, W.J. Gutowski, T. Johns, G. Krinner, M. Shongwe, C. Tebaldi, A.J. Weaver and M. Wehner, 2013: Long-term Climate Change: Projections, Commitments and Irreversibility. In: Climate Change 2013: The Physical Science Basis. Contribution of Working Group I to the Fifth Assessment Report of the Intergovernmental Panel on Climate Change [Stocker, T.F., D. Qin, G.-K. Plattner, M. Tignor, S.K. Allen,

J. Boschung, A. Nauels, Y. Xia, V. Bex and P.M. Midgley (eds.)]. Cambridge University Press, Cambridge, United Kingdom and New York, NY, USA.

Fitzsimmons, K. E., Cohen, T. J., Hesse, P. P., Jansen, J., Nanson, G. C., May, J.-H., Barrows, T. T., Haberlah, D., Hilgers, A., Kelly, T., Larsen, J., Lomax, J., and Treble, P.: Late Quaternary palaeoenvironmental change in the Australian drylands, Quaternary. Sci. Rev., 74, 78–96, 2013.

Flato, G., J. Marotzke, B. Abiodun, P. Braconnot, S.C. Chou, W. Collins, P. Cox, F. Driouech, S. Emori, V. Eyring, C. Forest, P. Gleckler, E. Guilyardi, C. Jakob, V. Kattsov, C. Reason and M. Rummukainen, 2013: Evaluation of Climate Models. In: Climate Change 2013: The Physical Science Basis. Contribution of Working Group I to the Fifth Assessment Report of the Intergovernmental Panel on Climate Change [Stocker, T.F., D. Qin, G.-K. Plattner, M. Tignor, S.K. Allen, J. Boschung, A. Nauels, Y. Xia, V. Bex and P.M. Midgley (eds.)]. Cambridge University Press, Cambridge, United Kingdom and New York, NY, USA.

Mauri, A., Davis, B. A. S., Collins and P. M., Kaplan, J. O., (2014), The influence of atmospheric circulation on the mid-Holocene climate of Europe: a data-model comparison, Clim. Past, 10, 1925–1938.

Petherick, L., Bostock, H., Cohen, T., Fitzsimmons, K., Tibby, J., Fletcher, M.-S., Moss, P., Reeves, J., Mooney, S., Barrows, T., Kemp, J., Jansen, J., Nanson, G., and Dosseto, A.: Climatic records over the past 30 ka from temperate Australia – a synthesis from the Oz-INTIMATE workgroup, Quaternary. Sci. Rev., 74, 58–77, 2013.

Prado, L. F., Wainer, I. and Chiessi, C. M., (2013), Mid-Holocene PMIP3/CMIP5 model results: Intercomparison for the South American monsoon system, The Holocene, 23, 1915–1920.

Reeves, J. M., Barrows, T. T., Cohen, T. J., Kiem, A. S., Bostock, H. C., Fitzsimmons, K. E., Jansen, J. D., Kemp, J., Krause, C., Petherick, L., Phipps, S. J., and Members, O.

I.: Climate variability over the last 35,000 years recorded in marine and terrestrial archives in the Australian region: an OZ-INTIMATE compilation, Quaternary. Sci. Rev., 74, 21–34, 2013a.

Reeves, J. M., Bostock, H. C., Ayliffe, L. K., Barrows, T. T., Deckker, P. D., Devriendt, L. S., Dunbar, G. B., Drysdale, R. N., Fitzsimmons, K. E., Gagan, M. K., Griffiths, M. L., Haberle, S. G., Jansen, J. D., Krause, C., Lewis, S., McGregor, H. V., Mooney, S. D., Moss, P., Nanson, G. C., Purcell, A., and van der Kaars, S.: Palaeoenvironmental change in tropical Australasia over the last 30,000 years – a synthesis by the OZ-INTIMATE group, Quaternary. Sci. Rev., 74, 97–114, 2013b.

Suppiah, R. (1992), The Australian summer monsoon: A review, Prog. Phys. Geogr. 16, 283–318.

**Restructuring**

This part outlines the restructuring that we aim to do to answer several of the key points the reviewer raises above. The authors feel that is would enhance the paper and make it more focussed. The authors do feel however, that the sections removed be included as a "Supplementary Material" section. The main reasons for this are:

1. Such a synthesis study has not been undertaken before and so having the analysis attached to the paper (as a supplement) will be useful to allow others to repeat our investigation (and validate or verify our conclusions).
2. There are interesting features in the analysis (primarily related to the seasonal cycle and southern Ocean clouds/ocean processes) that may provide a suitable background from which to perform further analysis.

The authors would do the following:

A. Remove section 4.1.1 and Figure 6 from the main paper and put them in the supplementary material. We will also show the seasonal change in circulation (annual, warm and cold seasons) across the Tropical North West in the Supplementary Material to explicitly show the circulation

changes that are referred to from the old Section 4.1.1. We have included that plot at the bottom of this response to show what we intend to include.

B. Remove Section 4.1.4 from the main paper and put in the supplement. This section does not add much to the paper, as there are few proxy measurements to back up the model interpretation. We will either state this issue in the results or conclusions section and also refer to it in the future work section (i.e. to resolve things better there). We will suggest that any reader looks at the supplement for a short discussion though.

C. Remove the boxes relating to the Northern Southern Ocean region in Figure 10 and any reference to that region in the discussion. We will move the boxplots and references to the Northern Southern Ocean into the supplementary material and the results section.

These three parts of the discussion are primarily associated with the places our statistical interpretation is somewhat ambiguous. Nonetheless, the discussion given is still a useful source of information for subsequent users of PMIP2 and PMIP3 (and even PMIP4) data.

Regarding the results, the authors can see there is repetition and the idea was to quickly recap the results to lead into the discussion; however, we accept that this could be done differently (and more concisely) and we will address it. We intend to start each of the results sections with the proxy interpretation and then refer to the model estimates. That should focus the paper towards the places where we have adequate proxy coverage. In the other areas, we will state the temperature and precipitation changes if they are significant in the models and then refer to those places in the final conclusions section as regions to find more data. We would integrate the following into the results and remove most of the wording from the discussion:

A. Section 4.1.2 paragraphs 1 and 3 along with the first 3 sentences of paragraph 4.

B. Paragraph 2 in Section 4.1.3.

C. All references to the NSO region in 4.1.4.

The authors would keep the discussions in 4.2.1 and 4.2.2 almost unchanged as the points raised there are important for the overall ability of models to simulate important aspects of the climate system that are known to be poorly represented (and further verified in this paleoclimate synthesis). The authors will also endeavour to discuss only those features (temperature and precipitation) that are important and significant. Other features that may be interesting or useful for readers will be placed in the supplement.

The authors also propose to put the current Figures 4 and 5 in the supplementary material too and only state the seasonal values for temperature and precipitation (and model agreement) in the text if there are proxy data to compare with. We will also refer to these supplementary figures from the new results section and also in the future work section. This will allow us to highlight only the key regions that have seasonally resolved proxy data in order to showcase their importance for comparison with the model output. Removing these figures will make the paper more concise; however, we believe it is necessary to keep them in a supplement in order for others to view them if they wish to progress this work further.

Finally, with respect the to future work section, we will think bigger in terms of what should be done with the proxies and extend that section somewhat. As for the models, we will incorporate the references the reviewer suggests at the start of the review and include a discussion of those options too. We will also clarify our given suggestions by explaining why they should be undertaken (as described in the response to the reviewer's "Future directions" comment above).

[Figure]

Figure TNW: The multi-model mean circulation over the Tropical North West Domain for the 0 ka simulations' (a) annual mean, (b) October–March (WRM) season mean and (c) April–September (CLD) mean. Corresponding figures for the 6 ka simulations are plotted in (d)–(f) with the differences (d minus a, e minus b and f minus c) plotted in (g)–(i), respectively. The change from relatively moist northeasterlies to relatively dry southeasterlies can be seen from WRM to

CLD in both the 0 ka and 6 ka simulations. The slightly stronger northeasterlies in WRM (Figure (h)) are consistent with the slight increase in precipitation for TNW in October to March (as can be seen in the seasonal cycle plot, Figure 6 in the original version of the paper). The models are therefore consistent with the literature and our assessment. This figure will be included in the supplementary material too.

---

## Author Comment (AC2) · 14 Apr 2017

*I openly declare ahead of this review that I know almost all of the authors of this paper personally. I've worked directly with many of them in the past on other published work and with some of them on previously funded projects. I have no personal interest conflicts with any of them.*

*An additional review was requested of me by CPD because we have been unfortunate to have only one solicited review for this paper. As such I submit this review here, but would like the authors to understand that final adjudication will be undertaken in consultation with the editors of CPD.*

**Response:** Firstly the authors would like to thank Drew for undertaking the review of this manuscript, given the circumstances mentioned.

*Overall, I thought this was a useful manuscript to put forward and it will be a welcome addition to the regional literature when completed. Attempts to reconcile Australian palaeoclimate data with simulations are not numerous. The region has a great number of complexities of climate drivers and processes that could be evaluated using model proxy comparisons.*

**Response:** We agree that this region has largely been overlooked in global compilations and we hope this contribution goes some way to addressing this.

*However, I have some concerns about how this paper has been presented though, or at least how a few things are couched, and with some of the visual content that is presented (or data not shown). I suggest it could be publishable in CP, but only if many minor and some major revisions were undertaken. Please see specific comments in the attached PDF.*

**Response:** These are addressed individually below.

*First, the balance of the paper strengths lean heavily toward the analysis and descriptions of the climate model simulations. I'm elated that some basic physics of the climate system has been brought to bear with the incorporation of the discussion on the latitudinal gradient and geostrophic wind equations. There are some very interesting findings here for the model results, but I also cannot determine if any bias corrections were actually undertaken for the circulation patterns or if there is simply reference to identifying them using the post 1750CE data sets. Please make this clearer.*

**Response:** No bias corrections were undertaken on the circulation, precipitation or temperature patterns in the models. We are actually looking for those biases and therefore any correction applied to the model data would negate that. In reference to the cold tongue bias and the Southern Ocean temperature biases specifically, we use the post 1750 datasets to identify processes that have been described elsewhere in the literature (which has been cited); however, our study shows that:

1. The biases are present in the pre-industrial control runs (and not just the historical simulations, which are the focus of the cited literature).
2. The biases are present regardless of the time period under consideration, which is very clear when compared to the paleo-proxy reconstructions too.
3. Most important of all, the biases can actually be made worse in the 6 ka simulations as the changes in insolation appear to enhance those errors further.

We do state in section 2.3 that, "As neither of these datasets is representative of the climate at 1750 C.E. (as in the 0 ka simulations), they are only used to highlight known biases in the GCM simulations that may cause discrepancy relative to the proxies". The post 1750 CE datasets are therefore extra confirmation only, in order to draw attention to the processes that drive the well-known model biases. The authors therefore do not see how this can be made clearer than it already is in the paper.

*Ahead of the following comments, I would also suggest the authors simply recast the use of the palaeo proxy data network (from Reeves et al., 2013a) as supporting field-based information that the models can be compared against (details why are stated below). Section 2.1 is also poorly written; it leads off with a description of INTIMATE and then Reeves et al 2013 - and it seems very odd to me that for a SHAPE issue that there is no mention of*

*that initiative anywhere, which has superseded INTIMATE in the Southern Hemisphere. I think it would be more contemporary to refine the aims of the model-proxy intercomparison in light of the stated goals of the SHAPE IFG - which are similarly stated on the SHAPE project website.*

**Response:** The authors apologise for this oversight and this will be rectified in the revised manuscript with the following text:

"This research forms a contribution to the Southern Hemisphere Assessment of PalaeoEnvironment (SHAPE) program, an INQUA International Focus Group (project 1067P). SHAPE is the successor to the Aus-INTIMATE program, focussing on the Southern Hemisphere. One of the key remits of SHAPE is to develop model-proxy comparisons to both help understand the dynamic mechanisms behind regional palaeoclimate proxy-based reconstructions and test the robustness of the palaeoclimate models."

*Second, there appear to be no real surprises to me in terms of the findings – the proxy data-model comparisons are elementary (mostly descriptive, but still very useful and clearly-written). They are divided into sections that essentially show where the proxy-model comparisons work, and where they don't. I would greatly appreciate if the listing of proxy data derived from Reeves et al., 2013 (fundamental to supporting this work) was tabulated, including all metadata about location, type of archive, dating controls, seasonal sensitivity and signature for climate during the 6k interval are stated.*

**Response**: We agree that this would be a useful contribution and would be willing to undertake this. However, we would request an additional 3 months to allow someone to be employed to undertake this task. Much of this material is available in the regional papers, however having these accessible via database would be of benefit to the broader community.

*The Reeves et al., 2013 paper is also mentioned as providing 'a method' but it does not do that in terms of integrating the data or providing a dynamical understanding of past variability or change with reference to a mean climate state. That particular work collected climate proxy records under certain criterion, and binned them into different geographic*

*regions for Australia. If you adopt the spatial division of Reeves et al., 2013, and the data series used there in, it would be best to simply say so.*

**Response**: We will amend our text accordingly as the editor suggests.

*There in, those geographic regions are somewhat arbitrarily ascribed; but I temper this comment by saying in reality there is good reason to have made those divisions. Just a bit more support and justification from modern climate studies that indicate there is a strong reason for the geographic divisions would go a long way to informing the readership.*

*I believe that information can be easily obtained, and cited in the revised work. And better recognition that the real strength of Reeves et al., 2013 is the pre-selected proxy data that are 'regionally-representative'.*

**Response**: There is a substantial description of the modern regional climate provided in Reeves et al., 2013 and the regional-specific papers of the same volume. We would be reluctant to repeat too much of this, but can direct readers to this justification and the updated Köppen-Geiger classification system (Peel MC, Finlayson BL & McMahon TA (2007), Updated world map of the Köppen-Geiger climate classification, Hydrol. Earth Syst. Sci., 11, 1633-1644), which is broadly comparable. The actual delineation of boundaries (using straight lines) is somewhat arbitrary, however it is the clustering of records with the regions that is more robust. As mentioned in the response to the other reviewer (and to be added to the text) these records were selected by many of the Australian palaeoclimatic community as being the most regionally representative.

*Third, the Reeves et al., 2013 depictions did not compare the past climate change signals to a common modern interval, but rather assessed the direction of change from one time step to the next. This limits meaningful comparisons of the past patterns that are shown in Reeves et al., 2013 to the climate model simulations shown in this study.*

*I realise there were probably data limitations in Reeves et al., 2013 that sent those authors down such a path, but it was identified as problematic early on (in discussions in Aus-*

*INTIMATE). In this paper, it (and the pictures showing signals for different time slices) is advocated as 'presenting a new opportunity to integrate models with data'. At the risk of repeating myself, it does not: What it does is supply a series of pre-screened data and climate signals for the mid Holocene where assessing PMIP2 and 3 model signals may be undertaken. The authors have largely done this in a point-by-point fashion; if the data from Reeves et al. has been further transformed, it is not clear how it was done. Deeper understanding from data integration would have been more meaningful; so I feel justified in mentioning this specific point here.*

**Response**: We acknowledge that the Reeves et al., 2013 method of comparing directions of change is not to everybody's taste. However, this approach was applied to help understand the 'on-ground' transitions in climate, particularly from a biogeographic perspective. This manuscript has gone back to those records and focussed in on the 6ka timeslice, in comparison to pre-industrial modern, where possible. We will make this statement more clearly in the revised manuscript. Furthermore, we also believe this does provide an opportunity to integrate models with the proxy data for two reasons:

1. It provides a way to diagnose model output relative to the proxy archives over a spatial scale they can resolve (i.e. many grid points are used for the spatial averaging).

2. It provides a regional overview of the state of the climate that *could* be used in data assimilation i.e. it may be more useful to integrate the regional-scale temperature anomalies rather than individual point measurements, for example. This is something that could (and probably should) be attempted in the future for proxy system modelling.

While we do see the editor's point about this not being a model-proxy integration method (which it is not), there is definitely a case for this. Nonetheless, we will change the words "integration" to "comparison" within the text, as that is more appropriate language.

*Fourth, I would also strongly encourage the authors to submit the data from Reeves et al., 2013 along with this paper, or provide a supplement with stable URLs where the data may be obtained. Sub-issues related to the points of viewing and assessing those data are: a. mapping of proxy signals onto the PMIP simulation outputs shown in Figures 3, 4, and 5. b.*

*being able to observe the time series for each c. seeing how the 6k signatures compare to modern or pre industrial times.*

**Response**: As mentioned above, we agree this would be an important contribution, but require time to compile this. We would like to request 3 months. Nevertheless, the authors also feel that this request is getting beyond the scope of the paper. Such a process should really be considered as a long-term investment by the SHAPE initiative beyond just the mid-Holocene (and this study). Furthermore, the work is intended to provide an important, modelling perspective on mid-Holocene climate over Australasia. This paper therefore, already provides a timely modelling-centric synthesis, which has been a significant omission in the literature thus far. It also directly points the way to areas that could easily become the focus of future work.

*Fifth, the scaling of the proxy signals so that they are compatible with the GCM signals is still unclear to me. This relates to point number 3. In using a tercile-based evaluation system of the proxy data, one needs to create a distribution for the data, with reference to a common interval (also the same interval used in the control run for the model simulations), then establish what the thresholds are for the terciles to obtain meaningful signals (warm, wet, cold, dry etc.). That has not been clearly shown anywhere here ... and it cannot rely on antecedent work. Seeing the data and the new analysis are required for the descriptions of the proxies to be understood as factual.*

**Response:** The spatial coverage of quantifiable, high-resolution palaeoclimate records across the Australian continent is sadly lacking. Whilst recent efforts in the palaeoclimate community are seeking to address this, we can currently only work with the records we have. However, the records included in this compilation are those that have been selected by many of the Australian paleoclimate community to be most representative of the respective regions discussed. As noted above, what is attempted here is to compare the relative climatic change as described by the model and proxy records between 6 ka and modern. Even at this coarse scale (warmer, colder; wetter, drier), meaningful comparisons between the proxies and models have been made and clearly point to areas (both geographical and scientific) that warrant more targeted research. We are happy to make the

metadata available in supplementary material for future analysis; however, comparisons with peer reviewed, antecedent work are important and necessary steps for **any** piece of research and has been undertaken very carefully (and in-depth) in this study. Such comparisons must therefore remain in the paper otherwise all context for the work will be lost.

*Addressing the above comments, the more minor grammatical issues in the text, and recasting the paper toward the main strengths (modelling results and forcing mechanisms, supported by point data, rather than proxy-model intercomparison) would see this through. I'd also like to encourage the authors to evaluate their future work section and to try to be broader with regard to proxy development, chronology evaluation and integrative approaches that could help future efforts bring models and proxies together - please see if that can also be done in a more refined manuscript.*

**Response:** As per the response to Reviewer 1, yes we will recast the future work section, particularly with regards chronology and uncertainty of the proxy-based reconstructions. However, the regional model-proxy comparison is hugely important to this. Comparing point-for-point model and proxy data is nonsensical, as the models cannot resolve the necessary processes that occur at such a fine scale. Also, the proxies are clearly representative of a wider area than the individual records represent—and on scales the models can resolve (i.e. regionally). For example, a model grid box may cover 22,500 km$^2$ (i.e. 150 km x 150 km) over a topographically varied region, e.g. the Southern Alps. Within that box, there could be many individual proxy records that are airflow direction dependent (e.g. leeward versus windward slopes). Comparing such varied data within a GCM is nonsensical and impractical, as parameterization does not explicitly resolve such complexities. By looking at regionally coherent signals over multiple reconstructions (as done here) we begin to build up a picture of the overall climate throughout a region that the GCMs may be able to represent. Only at this point is it worth comparing the two datasets and not on a point-by-point basis as suggested. The strength of this work is the regional comparison (as Ackerley et al., 2011, did for their New Zealand study) and absolutely needs to be maintained as such. We will make sure a statement outlining this point is included in any revised paper, as we believe it is very important.

**Response to specific comments in manuscript (given as a supplement to A. Lorrey's review). Other minor comments will be addressed subject to the decision on whether we are asked to submit a revised manuscript.**

Note: in general we accept the grammatical and stylistic suggestions put forward and are happy to incorporate most of them in an updated version of the paper.

**P1l1** We will tone down the references to INTIMATE and restructure in view of the contribution to SHAPE – particularly at the outset of this abstract.

**P2L16** Yes – we will redefine the Southern Ocean to be in keeping with the area in reference.

**P2L18** – Regional divisions and temporal reference addressed above. We suggest modifying the text to read:

Over Australasia, the Maritime Continent and Southern Ocean, such an upscaling/downscaling approach has not yet been attempted to integrate proxy and model data; however, the synopsis of the OZ-INTIMATE initiative Reeves et al. (2013a) presents an opportunity to do so, which is explored here. In particular, there is the move beyond the descriptive and begin to test some of the dynamical mechanisms represented by the proxy response, providing a deeper understanding of some of the key drivers of change.

**P2L24** – Addressed above

**P2L30** - Addressed above

**P4L7** – Addressed above – the 6k palaeodata is compared to the present

**P4L10** – there are no watermelons – maybe "Granny Smiths vs. Gravensteins", but we understand the point being raised fully. We will make it clearer in the text that this paper is

comparing 6k to 0k, using the INTIMATE suite of papers, but not the same approach (that is trends through time). This contribution uses the SHAPE approach of comparing past conditions to modern conditions.

**P6L12 –** The 1870–1899 data were used, as they were the closest we could get to observational data from actual instruments. Furthermore, we wanted as much of the modern-day (and overall 20[th] Century) warming signal to be removed so as not to exacerbate the already existing cold tongue bias (with respect to the Tropical Pacific). We know the bias is there (abundantly clear in the literature) and we wanted to use some reference dataset to give it context, in this case we chose HadISST. We are reluctant to extend the averaging period as it could artificially make the biases appear to be worse than they are due to recent warming. Nevertheless, we will state in the paper that we used this time period to minimise the impact of climate change on the temperature record. N.B. when undertaking our proposed analysis with responses to P23L15 and P26L16 below, if it becomes apparent that we should mention the use of more modern SSTs then we will also include it in Section 2.3.

**P9L17** - As per comment above, we agree that tabulating the source data and making it available through supplementary material would be a valuable contribution, if we can be permitted the extra time this would entail to compile. In addition, we can provide a simple annotation to the maps provided to indicate direction of change (see example Figure EX1 below). We will also re-structure the wording to make sure that it is consistent with the figure (and adjust the figure further if necessary—the Fig. EX1 below is just an initial draft for the purposes of this response).

**P23L15 –** This is a good point. While there were likely to have been ship observations across the Pacific during 1870 to 1899 (and land-based observations to infer the polarity of the Southern Oscillation Index and an inference of El Nino/La Nina), it is unlikely that such observations would have been taken over the Southern Ocean. We will therefore include the value for the modern (1980 to 2009) SSTs in this region and include the reasoning behind this here. Again, given the Southern Ocean cloud and temperature biases are large

(and discussed in-depth in the cited literature) it is unlikely to alter our conclusions but we fully agree that this is a necessary checkpoint here.

**P26L16 –** We completely see the "incompatibility" argument here between using the 1870 to 1899 HadISST dataset and the 1979 to 2008 ERA-Interim dataset. The reasons for this are:

1. To avoid overemphasising the cold tongue bias by including the climate change signal in the 1979 – 2008 period.
2. ERA-Interim (and therefore any measure of the circulation over the Pacific) does not run back to 1870 – 1899 and so it is our only observationally constrained estimate of the mean circulation.

It is important to note that we are not presenting a case for a new process, merely highlighting that something that is known as an important modelling problem is also seen in the mid-Holocene simulations. Furthermore, the error is made worse in the 6 ka simulations because of the error in the base state, which is clearly important for any 6 ka GCM study. That said, we would be willing to include a figure in the supplementary material showing the current Figure 11 with SST and circulation from the same reference time period, to validate the editor's point. We would then state in the text that using the modern period does not change the overall result.

**P30L24 -** We see that the use of a tool such as PICT could provide a positive way forward – but is outside the scope of this publication. The issues with MAT in Australia are particularly fraught with the uncertain and varied impact of a long human presence in the landscape, bringing into question some of the assumptions of climatic stationarity – even during the Holocene.

**P32L9** - this has been addressed above and will be more clearly detailed in the text.

[Figure]

Figure EX1: The ensemble and regional annual mean differences in (a) surface temperature (K), (b) precipitation (mm day−1 and [%]) and (c) 850 hPa circulation (m s−1) for the 6 ka simulations relative to the 0 ka simulations. In (a) blue shading represents lower area-averaged surface temperature and red indicates higher at 6 ka. Red circles indicate higher, blue circles indicate lower and grey circles indicate little change for temperatures at 6 ka relative to 0 ka from proxy estimates. In (b), orange indicates lower area averaged precipitation and green indicates higher at 6 ka. In both (a) and (b) the values of the ensemble mean changes are given in white and the percentages of models that agree on the sign (positive or negative) of the ensemble mean temperature or precipitation differences are given by the white numbers. Amber circles indicate higher, green circles indicate lower and grey circles indicate little change for precipitation at 6 ka relative to 0 ka from proxy estimates. White circles with an X inside indicate not data available (both temperature and precipitation). In (c) shading indicates the direction and strength of the ensemble mean 850 hPa zonal wind (blue colours = easterly and red colours = westerly) in the 0 ka simulations. This figure is purely an example here and will be refined and checked for any revised version of the paper.

---

## Author Response (AR1)

Dear Editor (Andrew Lorrey),

Please find our updated version of the manuscript entitled "*Evaluation of PMIP2 and PMIP3 simulations of mid-Holocene climate in the Indo-Pacific, Australasian and Southern Ocean regions*" (manuscript I. D. cp-2016-136) that has been submitted to the *Climate of the Past* special issue on *Southern perspectives on climate and the environment from the Last Glacial Maximum through the Holocene: the Southern Hemisphere Assessment of PalaeoEnvironments (SHAPE) project.*

We (the authors) have completed the revisions we outlined during the review stage. This submission includes the main revised document (Ackerleyetal.pdf), the supplementary material (Ackerleyetal_Supplement.pdf) and another supplement (SHAPE_Database_020817.xlsx) that shows the latest version of the database for the proxy records used in the paper. The database is not quite complete, but we are updating and finalising it presently. We envisage that it will be complete by the time of full publication if the paper is accepted. We would also recommend that the paper not be published until the database is in such a state that it is considered complete by the Editor and the authors (e.g. if there are any unforseen delays in the interim, we will inform the Editor and *Climate of the Past* immediately and work out a course of action). Please note that we have included a track changed document (i.e. latexdiff as specified—document is Ackerleyetal_latexdiff.pdf); however, the changes are so substantial that the document is almost illegible. We have included Ackerleyet_Supplement.pdf, SHAPE_Database_020817.xlsx and Ackerleyetal_latexdiff.pdf in the compressed file – Ackerleyetal_All_Supplements.tar.

The authors hope that the revisions that have been made to the main article, the new supplementary material document and the new database have answered all of the initial review points raised in the discussion paper.

We look forward to receiving your responses.

Kind regards,

Duncan Ackerley (lead author).